# The Cost of Learning Under Multiple Change Points

**Tomer Gafni** [1]   **Garud Iyengar** [1]   **Assaf Zeevi** [1]

## Abstract

We consider an online learning problem in environments with multiple change points. In contrast to the single change point problem that is widely studied using classical "high confidence" detection schemes, the multiple change point environment presents new learning-theoretic and algorithmic challenges. Specifically, we show that classical methods may exhibit catastrophic failure (high regret) due to a phenomenon we refer to as *endogenous confounding*. To overcome this, we propose a new class of learning algorithms dubbed Anytime Tracking CUSUM (ATC). These are horizon-free online algorithms that implement a selective detection principle, balancing the need to ignore "small" (hard-to-detect) shifts, while reacting "quickly" to significant ones. We prove that the performance of a properly tuned ATC algorithm is nearly minimax-optimal; its regret is guaranteed to closely match a novel information-theoretic lower bound on the achievable performance of any learning algorithm in the multiple change point problem. Experiments on synthetic as well as real-world data validate the aforementioned theoretical findings.

## 1. Introduction

### 1.1. Background

Online learning studies sequential decision-making problems in which a learner observes a stream of noisy data and selects actions to optimize some objective (typically, minimize regret) (Cesa-Bianchi & Lugosi, 2006). In many real-world settings, the data-generating process is non-stationary and often manifests as distribution shifts at unknown points over the problem horizon (Hartland et al., 2006). A simple instance of this problem is where the mean undergoes abrupt changes; identifying the latter is the subject of a vast litera-

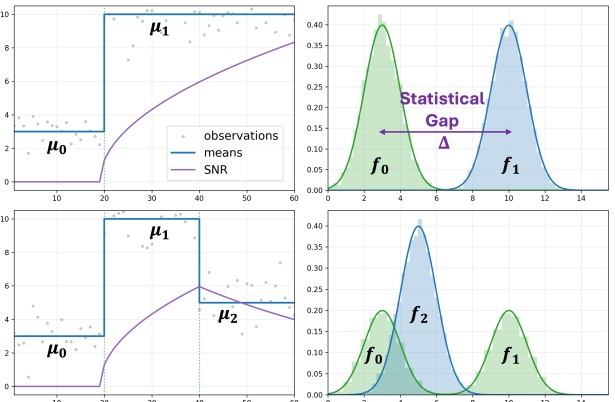

*Figure 1.* Effect of endogenous confounding on detection signal-to-noise ratio (SNR). Top: After a change at $t = 20$, accumulating post-change samples increases the SNR of the detection statistic. Bottom: After the second change at $t = 40$, failing to discard outdated samples from $f_0$ causes the reference statistic to be computed from a mixture of $f_0$ and $f_1$ (right), reducing statistical separability with respect to $\mu_2$, thereby degrading the SNR for detecting $f_2$.

ture on *change point detection* (Siegmund & Venkatraman, 1995; Liang & Veeravalli, 2022).

This paper adopts an online learning perspective to study the fundamental trade-offs in the multiple change point setting. While the single-change problem is well understood (Basseville et al., 1993), multiple changes introduce new difficulties, and existing extensions remain limited and typically rely on detectability assumptions such as minimum spacing between changes or minimum jump sizes (Maillard, 2019; Yu et al., 2023). Such difficulties are particularly pronounced in non-parametric settings, where the reference distribution used for testing is not known a priori and must be estimated from past samples.

To illustrate some of the challenges facing the learner in such problems, consider first the *single* change point setting. Standard learning and detection methods (e.g., $\delta$-PAC) focus on quickly identifying with high probability that a change has occurred; this is akin to the best arm identification under fixed confidence in the multi-armed bandit setting. When the pre- and post-change distributions are separated by a "statistical gap" (e.g., difference in mean) $\Delta$, the sample complexity to reach confidence level $1 - \delta$ typically scales as $\Delta^{-2} \log(1/\delta)$ (Garivier & Kaufmann, 2016).

[1]Columbia University, New York, NY, USA. Correspondence to: Tomer Gafni <tbg2124@columbia.edu>.

*Proceedings of the 43rd International Conference on Machine Learning*, Seoul, South Korea. PMLR 306, 2026. Copyright 2026 by the author(s).

A salient property of the single change point problem is that once a change has taken place, the post-change distribution remains fixed, so the more time "passes" after the change point has manifested, the greater the likelihood of identifying that the distribution has shifted. In contrast, in the multiple change point problem, if a detection algorithm fails to identify a change, then from that point on the reference distribution is no longer correctly specified. Consequently, additional data that is collected after said change-point may act as "noise", confounding the ability to detect the next change point. This artifact may cascade into subsequent detections, progressively degrading the power of the learning algorithm, and ultimately causing a breakdown in detection performance.

We illustrate this phenomenon in the mean-shift setting (Fig. 1). With a single change (top panel), samples collected before the change are informative as they provide a clean reference for distinguishing the post-change distribution $f_1$ from the pre-change distribution $f_0$, e.g., via estimating the mean $\mu_0$. However, after a change to $f_2$ (bottom panel), these same samples degrade detection: when incorporated into the reference statistics, they induce a mixture of $f_0$ and $f_1$ that reduces separability for detecting $f_2$. This constitutes a form of *endogenous confounding*: the learner's own failures to detect changes and discard outdated data determine the reference statistics used subsequently, possibly exacerbating the difficulty of detecting the next change.

To the best of our knowledge, (Yu et al., 2023) are among the first to point out that, absent highly restrictive assumptions, methods that focus on high-confidence detection do not extend meaningfully to multiple change-point settings. In the present paper we offer a slightly different perspective: rather than imposing such assumptions, we propose a learning-theoretic perspective that assigns a cost to incorrect detections, and evaluates performance via cumulative regret. The latter naturally encapsulates the fundamental costs of learning in the presence of multiple change points. This includes detection delays, which depend on the intrinsic difficulty of each shift, false alarms, and, importantly, the endogenous confounding induced by missed detections, a phenomenon that is ruled out by the high-confidence assumptions typically imposed in the change-point literature.

We study this model through an online tracking problem under a dynamic regret formulation (i.e., where the oracle dynamically tracks the true underlying statistical model at all points in time), which serves as a canonical setting for capturing these considerations. A learner sequentially observes noisy sub-Gaussian samples $\{X_t\}_{t=1}^{T}$ whose mean $\mu_t$ evolves in a piecewise-constant manner with $S$ change points over a horizon $T$. The mean sequence $\{\mu_t\}$ and the parameters $T$ and $S$ are *unknown* to the learner. At each time step, the learner predicts the mean $\hat{\mu}_t$, and per-

formance is measured by the expected cumulative squared error $\mathbb{E}\left[\sum_{t=2}^{T}(\hat{\mu}_t - \mu_t)^2\right]$, corresponding to dynamic regret with respect to the time-varying mean sequence (Jadbabaie et al., 2015). Extant work on learning under abrupt changes typically imposes detectability assumptions (Maillard, 2019; Yu et al., 2023; Huang & Veeravalli, 2025). In contrast, we impose no assumptions on minimum shift magnitude or spacing between change points; instead, we frame the problem in a *minimax* (worst-case) regret framework. In this setting, missed detections may be unavoidable, and the resulting confounding poses both analytical and algorithmic challenges.

This formulation arises in a wide range of applications, including demand tracking for real-time resource allocation in online platforms (e.g., ride-hailing), where abrupt shifts may be caused by exogenous environmental shocks (e.g., extreme weather events), multi-period control problems, and online data compression. We further discuss concrete application scenarios and extensions to higher-dimensional problems in later sections.

## 1.2. Main contributions

Minimizing dynamic regret in our setting requires managing the trade-off between *adaptivity* and *stability*: rapidly discarding past data after genuine shifts are detected reduces bias; but overly reactive behavior (discarding informative data) increases variance. Endogenous confounding adds a new dimension to this tension: decisions made under one regime influence the statistical landscape faced by future detection and estimation tasks. Our central insight is that *attempting to detect every change is neither necessary nor desirable.* We show that the regret cost associated with missed detections is not uniform - "small" mean shifts or shifts that last for a short duration do not incur a high cost. The challenge is to appropriately classify said changes. Our algorithm employs a time-varying detection threshold that effectively executes this classification: it detects sufficiently large shifts quickly and incurs minimal cost due to missed changes. In particular, changes whose magnitude remains below the statistical resolution induced by the uncertainty of the current reference estimate need not be detected, as their contribution to the regret remains controlled.

Our main contributions are summarized below.

**An anytime tracking algorithm.** We propose the *Anytime Tracking CUSUM (ATC)* algorithm (Section 3) that tracks the evolving mean, without knowledge of the horizon $T$ or the number of changes $S$. The key design feature here is a fully data-driven adaptive restart with a selective detection mechanism, which enables stability (few unnecessary restarts), adaptivity (fast detection when shifts are genuine), and robustness to missed detections.

**Endogenous confounding.** We quantify the cost of missed detections, causing outdated data to act as noise, and affecting future detection power and regret performance. The key to this analysis is bounding the induced reduction in signal to noise ratio (Proposition 3.1).

**Logarithmic upper bound on regret.** We prove a non-asymptotic upper bound on the dynamic regret (Theorem 4.1) showing that, even in worst-case environments where shifts may be statistically undetectable, the dynamic regret grows at most on the order of $O(\sigma^2(S+1)\log T)$ up to problem-dependent constants.

**A lower bound on achievable performance and minimax optimality.** We complement the upper bound with a novel information-theoretic lower bound (Theorem 4.2) of order $\Omega\big(\sigma^2(S+1)\log(T/(S+1))\big)$, showing that ATC is nearly minimax optimal.

We validate the theory with extensive experiments on both synthetic and real data (Section 5).

### 1.3. Related literature

We discuss related work from two complementary perspectives.

**Online learning in non-stationary environments.** Early formulations of non-stationary online learning were developed under adversarial models, where performance is measured relative to the best fixed comparator in hindsight (Auer et al., 2002; Zinkevich, 2003). In many cases, the dynamic regret is a stronger performance criterion (Jadbabaie et al., 2015) and has been extensively considered in recent years (Zhao et al., 2024; Qian et al., 2024), with squared loss naturally arising, as in online non-parametric regression (Baby & Wang, 2019). Without additional structure, obtaining a bound on dynamic regret is not possible; however, with suitable regularity of the comparator sequence (Besbes et al., 2015), sublinear regret guarantees are achievable. A widely studied form of regularity is piecewise stationarity, which provides a structured statistical observation model and has been investigated in various online learning formulations (Chen et al., 2022; Hou et al., 2024). Huang & Wang (2025) develop an adaptive window-selection principle for statistical learning under gradual non-stationarity via quasi-stationary segmentation. In contrast, our focus is on abrupt multiple change points and the effect of missed detections in restart-based procedures. Another particularly relevant line of work is multi-armed bandits in piecewise-stationary environments, where the learner must also account for partial feedback and action selection. Existing approaches in this setting typically fall into two broad classes. *Passively adaptive strategies* discard past observations through discounting or sliding-window schemes (Garivier & Moulines, 2011). *Active adaptive strategies* combine a base learner with ex-

plicit restarts triggered by an online change-detection module (Liu et al., 2018; Cao et al., 2019; Besson et al., 2022). These methods rely on high-probability detection guarantees, typically derived under detectability assumptions, as the key ingredient in the regret analysis. Our formulation isolates the statistical change-point component and studies performance in the absence of any detectability assumptions. This perspective may help inform the design of active adaptive methods for piecewise-stationary bandits without such assumptions.

**Quickest change detection (QCD).** QCD studies abrupt distribution shifts with the goal of minimizing detection delay under constraints on false-alarm probability. Classical foundations go back to Page's cumulative-sum (CUSUM) procedure (Page, 1954), with minimax formulations introduced in (Lorden, 1971); see (Poor & Hadjiliadis, 2009; Xie et al., 2021) for surveys and monographs. Much of the literature addresses a single change point under a known pre-change distribution, with extensions allowing unknown pre- and post-change distributions (Lai & Xing, 2010). Computationally efficient CUSUM-type procedures have also been developed for unknown post-change parameters, e.g., using sliding-window estimates within recursive CUSUM updates (Xie et al., 2023). Extensions to multiple change points are studied in (Maillard, 2019; Yu et al., 2023). These works also rely on separability conditions to guarantee detectability of each change. To the best of our knowledge, a minimax formulation of online QCD with multiple change points, without detectability assumptions, has not been studied. Our procedure is also distinct from joint detection-and-estimation formulations (Moustakides et al., 2012; Chen et al., 2013), where estimation of a parameter or state is part of the terminal inferential task. In contrast, we use the squared tracking error criterion to quantify the learning cost induced by detection delays, missed detections, and confounding.

We finally note a vast literature on offline multiple change-point detection, as well as Bayesian and Markovian formulations of related problems. While these works share conceptual similarities in modeling regime changes, their settings, analytical goals, and methodological approaches differ from those considered here.

## 2. Problem formulation

We study online tracking of a system mean parameter that undergoes abrupt shifts over time. The learner observes a sequence of real-valued random variables $\{X_1, \ldots, X_T\}$. The mean shifts are induced by an unknown deterministic sequence of $S$ change points $1 = \tau_0 < \tau_1 < \cdots < \tau_S < \tau_{S+1} = T + 1$, such that the observations $\{X_t\}_{t=1}^T$ are independent, and for each $j \in \{0, \ldots, S\}$, the segment $\mathcal{I}_j := [\tau_j, \tau_{j+1})$ consists of independent and identically

distributed samples with mean $\mathbb{E}[X_t] = \mu_j$, where $\mu_j$ is unknown. For convenience, we define the time-varying mean $\mu_t := \mu_j$ whenever $t \in \mathcal{I}_j$, so that $\mu_t = \mathbb{E}[X_t]$ for all $t$. We assume that the centered variables $(X_t - \mu_t)$ are sub-Gaussian with known variance proxy $\sigma^2$ [1]. As is standard in online learning to ensure finite regret bounds (Hazan et al., 2007), we assume that there exists an unknown constant $M \in (0, \infty)$ such that

$$\Delta_{\text{diam}} := \max_{0 \leq u, v \leq S} |\mu_u - \mu_v| \leq M. \tag{1}$$

Figure 2 (top) illustrates a typical environment of this form.

Let $\mathcal{F}_t := \sigma(X_1, \ldots, X_t)$ be the natural filtration, and let $\Pi$ denote a class of (possibly randomized) policies, where each $\pi \in \Pi$ outputs a prediction $\hat{\mu}_t^\pi$ at time $t$ that is measurable with respect to $\mathcal{F}_{t-1}$ (and the policy's internal randomness). An environment $\nu$ is specified by the change points and segment means, i.e., $\nu = (\{\tau_j\}_{j=1}^S, \{\mu_j\}_{j=0}^S)$, and we write $\mathbb{E}_{\pi,\nu}$ for expectation under this data-generating process and the policy's internal randomness. We evaluate $\pi$ in environment $\nu$ by its expected cumulative squared error (a dynamic regret criterion):

$$\mathcal{R}_T(\pi, \nu) = \mathbb{E}_{\pi,\nu}\left[\sum_{t=2}^T (\hat{\mu}_t^\pi - \mu_t)^2\right]. \tag{2}$$

For a policy $\pi \in \Pi$ and an environment class $\mathcal{E}$, define the worst-case regret $\mathcal{R}_T(\pi; \mathcal{E}) := \sup_{\nu \in \mathcal{E}} \mathcal{R}_T(\pi, \nu)$. The *minimax* regret over $\mathcal{E}$ is

$$\mathcal{R}_T^\star(\mathcal{E}) := \inf_{\pi \in \Pi} \mathcal{R}_T(\pi; \mathcal{E}) = \inf_{\pi \in \Pi} \sup_{\nu \in \mathcal{E}} \mathcal{R}_T(\pi, \nu). \tag{3}$$

Let $\mathcal{E}_{S,T}(\sigma^2, M)$ denote the class of piecewise-stationary mean environments with at most $S$ change points over horizon $T$, sub-Gaussian noise with proxy $\sigma^2$, and satisfying Eq. (1). Throughout, we take $\mathcal{E} = \mathcal{E}_{S,T}(\sigma^2, M)$.

## 2.1. Illustrative Examples of the Model

The piecewise-stationary mean-tracking formulation arises in a broad range of domains, including large-scale online platforms (e.g., ride-hailing, food delivery, and e-commerce) where demand is tracked for real-time resource allocation (Chen et al., 2021), and data-center resource monitoring (Maghakian et al., 2019). Closely related problems also arise in other research domains, such as universal compression of piecewise-stationary sources (Shamir & Merhav, 2002) and adaptive design in multiperiod control (Lai & Robbins, 1979; Jedra & Proutiere, 2022). We next present two illustrative examples.

**Example 1 (Ride-hailing).** Consider a ride-hailing platform (e.g., Uber/Lyft) that tracks local demand in a fixed geographic zone over time slots $t = 1, \ldots, T$ (e.g., 5-minute

intervals) to support real-time driver positioning and dispatch. Let $\lambda_t$ denote the latent request intensity (demand rate) in that zone, which can change over time due to exogenous factors such as weather, public events, or transit disruptions (Xu et al., 2025). A standard abstraction is that $\lambda_t$ is piecewise constant with unknown change points $\{\tau_j\}$.

In time slot $t$ (of length $\delta$), let $N_t$ be the number of incoming ride requests and define the observation $X_t := N_t/\delta$. Then $\mathbb{E}[X_t] = \lambda_t$, so identifying $\mu_t \equiv \lambda_t$ reduces demand tracking to our mean-tracking formulation. Operationally, the platform uses an online estimate $\hat{\lambda}_t$ to set supply targets (e.g., how many drivers to reposition into the zone). Under smoothness/curvature conditions on a convex mismatch cost that penalizes under- and over-supply (e.g., via waiting time, lost requests, and driver idle time), the incremental performance loss from using $\hat{\lambda}_t$ instead of $\lambda_t$ is locally quadratic in the estimation error and therefore scales as $(\hat{\lambda}_t - \lambda_t)^2$. Consequently, $\mathbb{E}[\sum_{t=2}^T (\hat{\lambda}_t - \lambda_t)^2]$ serves as a meaningful proxy for cumulative operational loss.

**Example 2 (Piecewise-stationary source coding).** Consider a streaming compressor that losslessly encodes a binary sequence $X_t \in \{0, 1\}$ whose distribution changes abruptly, e.g., a sensor toggling between operating modes (Chittam et al., 2018). In each stationary segment, symbols are independent with $\mathbb{P}(X_t = 1) = \theta_t$, where the parameter $\theta_t$ is piecewise constant with unknown change points. A strongly sequential encoder (e.g., arithmetic coding with online probability assignment) maintains an estimate $\hat{\theta}_t$ and incurs the per-symbol ideal codelength: $\ell_t = -\log(\hat{\theta}_t^{X_t}(1 - \hat{\theta}_t)^{1-X_t})$. The expected excess codelength relative to the true (unknown) model equals the Kullback–Leibler (KL) divergence between two Bernoulli distributions with success probabilities $\theta_t$ and $\hat{\theta}_t$, which is locally quadratic in $|\hat{\theta}_t - \theta_t|$ when $\theta_t$ is bounded away from 0 and 1. Thus, controlling $\mathbb{E}[\sum_{t=2}^T (\hat{\theta}_t - \theta_t)^2]$ serves as a meaningful proxy for controlling cumulative redundancy. Missing a change causes the estimator to mix samples across segments, resulting in a mismatched probability model and degraded compression performance.

# 3. The Anytime Tracking CUSUM (ATC) algorithm

In this section, we present the Anytime Tracking CUSUM (ATC) algorithm. ATC is an anytime online algorithm that only requires knowledge of the sub-Gaussian proxy $\sigma$. In particular, it does *not* require prior knowledge of the horizon $T$ or the number of change points $S$. We describe the structure of ATC in Section 3.1, analyze the stability–adaptivity trade-off in Section 3.2, and quantify the endogenous confounding effect of missed detections in Section 3.3.

---

[1] That is, there exists a known constant $\sigma^2 > 0$ such that, for all $t \geq 1$ and all $\lambda \in \mathbb{R}$, $\mathbb{E}[e^{\lambda(X_t - \mu_t)}] \leq \exp\left(\frac{\sigma^2 \lambda^2}{2}\right)$.

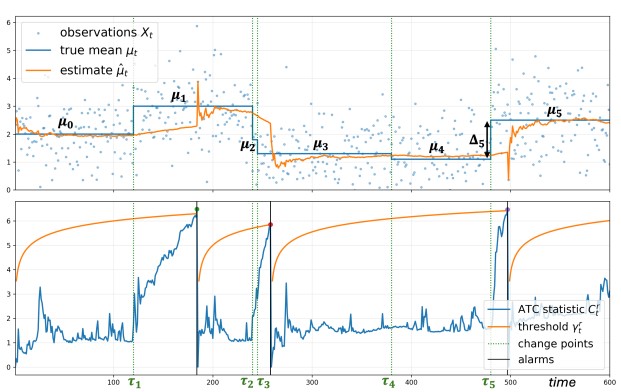

*Figure 2.* Example of an online tracking instance with multiple change points. The top panel shows the underlying piecewise-stationary environment, while the bottom panel illustrates the evolution of the detection statistic, the decision thresholds, and the alarms raised by the algorithm.

## 3.1. Algorithm structure

**Detection phase:** ATC uses a CUSUM-style statistic to determine whether a change has occurred. At time $t$, the algorithm has access to observations up to time $t-1$ and maintains a restart time $r < t$, corresponding to the most recent detected change. For each "split point" $k$ with $r < k < t$, define the two block averages

$$\bar{X}_{r:k-1} \;:=\; \frac{1}{k-r} \sum_{i=r}^{k-1} X_i, \qquad \bar{X}_{k:t-1} \;:=\; \frac{1}{t-k} \sum_{i=k}^{t-1} X_i.$$

Inspired by generalized likelihood ratio (GLR) tests for unknown means (Lai & Xing, 2010), we use the standardized two-sample mean-difference statistic

$$\hat{D}_{k,t}^r \;:=\; \frac{1}{\sigma} \sqrt{\frac{(k-r)(t-k)}{(t-r)}} \left| \bar{X}_{r:k-1} - \bar{X}_{k:t-1} \right|, \quad (4)$$

to define the *ATC statistic*

$$C_t^r \;:=\; \max_{r<k<t} \hat{D}_{k,t}^r. \quad (5)$$

Intuitively, $\hat{D}_{k,t}^r$ compares samples before and after a candidate split point $k$, and the scan over $k$ searches for the strongest evidence of a change since the last restart.

The corresponding *ATC test* that determines the alarm and restart time (stopping time) is

$$N_G^r \;:=\; \inf \left\{ t > r : C_t^r \geq \gamma_t^r \right\}, \quad (6)$$

with the time-varying threshold

$$\gamma_t^r \;:=\; \sqrt{6\log(t-r) + 2\log(1/\alpha_r) + 2\log(\pi^2/3)}, \quad (7)$$

where $\alpha_r := \frac{6}{\pi^2} \frac{\alpha}{r^2}$ for $\alpha \in (0,1)$ [2]. The term $6\log(t-r)$ provides uniform control over the scan statistic across all

---

[2] $\alpha$ is a user-specified tuning parameter. Our logarithmic regret bound holds for any fixed $\alpha \in (0,1)$ independent of $T$ and $S$.

times $t$ and split points $k$ within a fixed restart interval, while the summable allocation $\alpha_r$ ensures that this control holds uniformly over the possible restart times of the algorithm. After an alarm at time $t$, ATC restarts by setting $r \leftarrow t-1$.

**Prediction phase.** At each time $t > r$, ATC outputs the running average since the last restart $r$,

$$\hat{\mu}_t^r \;=\; \frac{1}{t-r} \sum_{i=r}^{t-1} X_i. \quad (8)$$

Ideally, we want an alarm to occur immediately after each mean shift to eliminate bias from outdated samples, while no alarms should occur during stationary periods, allowing the algorithm to retain all data and thereby reduce variance.

The detection statistic of ATC can be viewed as a generalization of the classical CUSUM procedure (Lorden, 1971) to a fully online setting with multiple change points and with unknown pre- and post-change distributions. Algorithm 1 summarizes the proposed procedure, and Figure 2 illustrates a representative run of ATC. The bottom panel plots the detector $C_t^r$ together with the threshold $\gamma_t^r$. Detected changes correspond to sharp increases of $C_t^r$ that cross the threshold, after which $\hat{\mu}_t^r$ rapidly re-centers around the new mean. In contrast, some changes do not trigger a restart (missed detections), reflecting ATC's selective detection principle.

## 3.2. Balancing the stability-adaptivity trade-off

ATC raises an alarm when the scan statistic $C_t^r$ exceeds the time-varying threshold $\gamma_t^r$. To understand how this rule balances stability and adaptivity, it is helpful to consider the population analogue of the two-sample statistic:

$$D_{k,t}^r \;:=\; \frac{1}{\sigma} \sqrt{\frac{(k-r)(t-k)}{(t-r)}} \left| \bar{\mu}_{r:k-1} - \bar{\mu}_{k:t-1} \right|, \quad (9)$$

where $\bar{\mu}_{m,n} := \frac{1}{n-m+1} \sum_{i=m}^{n} \mu_i$. We define the SNR governing detection as the population analogue of $C_t^r$:

$$\mathrm{SNR}(t;r) \;:=\; \left( \max_{r<k<t} D_{k,t}^r \right)^2. \quad (10)$$

This quantity characterizes the power of the ATC test at time $t$, given the last restart time $r$. In the Gaussian case, it is closely related to the KL divergence.

**Stability.** When no change occurs during $[r,t)$, the mean is constant and $D_{k,t}^r = 0$ for all $k$. The logarithmic growth of $\gamma_t^r$ guarantees uniform concentration of $\hat{D}_{k,t}^r$ around its population value over all split points $k$, ensuring that false alarms, and hence unnecessary restarts, occur with low probability. This behavior is illustrated by stationary segments in Figure 2, where $C_t^r$ fluctuates below the threshold.

**Adaptivity.** Consider a change at time $\tau_j$. In the ideal case where the previous change was detected and the algorithm restarted at $\tau_{j-1}$ (matching the single-change setting), the SNR is attained at the aligned split $k^\star = \tau_j$ and equals

$$\mathrm{SNR}_j^\star(t) := \mathrm{SNR}_j(t; \tau_{j-1}) = \frac{(\tau_j - \tau_{j-1})(t - \tau_j)}{t - \tau_{j-1}} \frac{\Delta_j^2}{\sigma^2}, \tag{11}$$

with $\Delta_j := |\mu_j - \mu_{j-1}|$. As expected, Eq. (11) implies that larger shifts induce higher SNR. As post-change samples accumulate, the SNR increases, allowing sufficiently large shifts to cross the threshold within a logarithmic delay. At the same time, ATC remains *selective* in the sense that it does not require every shift to be detected before the next change: shifts whose accumulated evidence remains below the threshold may remain undetected to preserve stability, and their effect is accounted for through the confounding analysis discussed below. This behavior is visible in Figure 2, where $C_t^r$ rises sharply and triggers a restart for large shifts, while smaller or short-lived shifts, such as those at $\tau_2$ and $\tau_4$, are missed.

### 3.3. SNR degradation due to missed detection

When a shift is missed, ATC does not restart, and subsequent detection statistics are computed using data spanning multiple regimes. As a result, future tests are no longer performed against the nominal pre-change mean, but against an *effective pre-change mean* formed by a mixture of past regimes (Figure 1). Concretely, suppose the $j$-th change occurs at time $\tau_j$. We are at time $t > \tau_j$, and the preceding changes $\tau_i, \dots, \tau_{j-1}$ were missed. In this case, the segment since the last restart (say segment $i$) contains multiple stationary regimes with means $\mu_i, \dots, \mu_{j-1}$ and corresponding lengths $n_i, \dots, n_{j-1}$. The reference pre-change mean used by the test is the length-weighted average,

$$\mu_{\mathrm{pre}}^{\mathrm{eff}}(r, j) := \frac{\sum_{\ell=i}^{j-1} n_\ell \mu_\ell}{\sum_{\ell=i}^{j-1} n_\ell}. \tag{12}$$

Detection of $\tau_j$ is now governed by an effective mean gap $\Delta_j^{\mathrm{eff}}(r) := \left| \mu_{\mathrm{pre}}^{\mathrm{eff}}(r, j) - \mu_j \right|$. This gap may be substantially smaller than the nominal gap obtained if outdated samples were discarded. To quantify this effect, define the corresponding effective SNR for detecting $\tau_j$ at time $t$ as

$$\mathrm{SNR}_j^{\mathrm{eff}}(t; r) = \frac{(\tau_j - r)(t - \tau_j)}{t - r} \frac{(\Delta_j^{\mathrm{eff}}(r))^2}{\sigma^2}. \tag{13}$$

Recall the example in Figure 1. The gap $\Delta_2$ between $\mu_2$ and $\mu_1$ is large and easily detectable. However, missing the $\mu_0 \to \mu_1$ change substantially reduces the effective gap $\Delta_2^{\mathrm{eff}}$ since the reference becomes a mixture of $\mu_0$ and $\mu_1$. This leads to a significant SNR degradation, and may cause the shift to $\mu_2$ to be missed, even though it would be

detected had $\mu_1$ been identified. This SNR reduction may further cause cascading missed detections. The following proposition shows that, under ATC, the SNR reduction is controlled.

**Proposition 3.1** (*Controlled SNR degradation*). *Fix a deterministic restart index $r$. Then there exists a universal constant $C > 0$ such that, with probability at least $1 - \alpha_r$, the following holds for every change index $j \in \{1, \dots, S\}$ satisfying $r < \tau_{j-1} < \tau_j$, and every $t \in \{\tau_j + 1, \dots, \tau_{j+1} - 1\}$: if the ATC detector has not raised an alarm by time $t$, i.e., $N_G^r > t$, then*

$$\left( \mathrm{SNR}_j^\star(t) - \mathrm{SNR}_j^{\mathrm{eff}}(t; r) \right)_+ \leq C \log\left( \frac{\tau_j - r + 1}{\alpha_r} \right). \tag{14}$$

The proof is given in Appendix A.6. Proposition 3.1 shows that, under the ATC detection rule, the degradation in effective SNR caused by outdated samples remains controlled, growing at most logarithmically with the elapsed time since the last restart. Intuitively, persistent contamination can occur only when the statistical evidence for the missed change is itself weak. Consequently, the effective SNR cannot deteriorate arbitrarily fast. This property is central to the analysis, since it implies that ATC can still "recover" and detect subsequent changes.

**Computational considerations.** ATC requires evaluating the test for each candidate split $k \in \{r+1, \dots, t-1\}$, i.e., on the order of $(t - r)$ candidates per time step. With standard maintenance of cumulative (prefix) sums over the current segment, each candidate evaluation can be carried out in $O(1)$ time, yielding $O(t - r)$ per-step complexity. To improve computational efficiency in practice, the maximization can be accelerated by restricting $k$ to a multiscale grid (Lai & Xing, 2010). For ATC, we consider geometric offsets $d \in \{1, \lceil b \rceil, \lceil b^2 \rceil, \dots\}, b > 1$ and include the candidates $k = r + d$ and $k = t - d$, yielding at most $2\lceil \log_b(t - r) \rceil + 1 = O(\log(t - r))$ candidates per time step. As indicated in our simulations, such an approximation preserves the logarithmic scaling of ATC, though it affects finite-sample behavior. A systematic study of computationally optimized algorithms is left for future work.

**High-dimensional extension.** ATC extends to vector-valued observations $X_t \in \mathbb{R}^d$ by replacing scalar averages with vector averages and replacing the absolute mean-difference in the statistic by an $\ell_2$-norm: $\hat{D}_{k,t}^r = \frac{1}{\sigma}\sqrt{\frac{(k-r)(t-k)}{t-r}} \|\bar{X}_{r:k-1} - \bar{X}_{k:t-1}\|_2$. The restart logic is unchanged, but each candidate evaluation now costs $O(d)$ time. The anytime threshold is identical to the scalar case, up to an additional $\sqrt{d}$ term accounting for high-dimensional concentration. Full details are given in App. A.7.

---

**Algorithm 1** Anytime Tracking CUSUM (ATC)

---

1: **Inputs:** sub-Gaussian proxy $\sigma$, error budget $\alpha \in (0, 1)$
2: $G_0 \leftarrow 0$; observe $X_1$; $G_1 \leftarrow X_1$
3: $r \leftarrow 1$; $\alpha_r \leftarrow \frac{6\alpha}{\pi^2 r^2}$
4: **for** $t = 2, 3, \ldots$ **do**
5:    **if** $t \geq r + 2$ **then**
6:       $\gamma_t^r \leftarrow \sqrt{6 \log(t - r) + 2 \log(\frac{1}{\alpha_r}) + 2 \log(\frac{\pi^2}{3})}$
7:       $\hat{D}_{k,t}^r \leftarrow \sqrt{\frac{(k-r)(t-k)}{\sigma^2(t-r)}} \left| \frac{G_{k-1} - G_{r-1}}{k-r} - \frac{G_{t-1} - G_{k-1}}{t-k} \right|$
8:       $C_t^r \leftarrow \max_{r < k < t} \hat{D}_{k,t}^r$
9:       **if** $C_t^r \geq \gamma_t^r$ **then**
10:          $r \leftarrow t - 1$; $\alpha_r \leftarrow \frac{6\alpha}{\pi^2 r^2}$
11:       **end if**
12:    **end if**
13:    **predict** $\hat{\mu}_t \leftarrow \frac{G_{t-1} - G_{r-1}}{t - r}$
14:    observe $X_t$; $G_t \leftarrow G_{t-1} + X_t$
15: **end for**

---

## 4. Regret Analysis

### 4.1. Upper bound

**Theorem 4.1** (*Regret upper bound for ATC*). *Fix $\alpha \in (0, 1)$. For any environment $\nu \in \mathcal{E}_{S,T}(\sigma^2, M)$, the regret of ATC, given in Algorithm 1, satisfies*

$$
\mathcal{R}_T^{ATC} \leq C_V \, \sigma^2 (S + 1 + \alpha)(1 + \log T) \\
+ C_B \, \sigma^2 S \log(T/\alpha) + C_M M^2 (\alpha + S), \quad (15)
$$

*where $C_V, C_B$ and $C_M$ are universal positive constants.*

The proof is given in App. A.3. A novel aspect of the analysis lies in its treatment of missed detections: rather than ruling them out, the analysis exploits the characterization of the resulting SNR degradation, formalized in Proposition 3.1, to control the induced contamination. For fixed $\alpha \in (0, 1)$ and $M$, Th. 4.1 implies $\mathcal{R}_T^{ATC} = O\left(\sigma^2(S + 1)\log T + M^2 S\right)$.

**Proof sketch.** The proof is based on five lemmas in App. A.3; we summarize the main ideas.

**Step 1: Regret decomposition.** Lemma A.1 shows that the dynamic regret decomposes as $\mathcal{R}_T^{ATC} = \mathcal{R}_T^{var} + \mathcal{R}_T^{bias}$, where $\mathcal{R}_T^{var}$ corresponds to the cumulative estimation variance and $\mathcal{R}_T^{bias}$ corresponds to the cumulative bias incurred after a change and only until the next alarm.

**Step 2: Confidence bound.** Lemma A.2 establishes that, for each deterministic restart index $r$, the deviation $|\hat{D}_{k,t}^r - D_{k,t}^r|$ is uniformly controlled over all split points $k$ and times $t$ with total budget $\alpha_r$. Since the restart times generated by ATC are random, whenever a deviation event is evaluated at a realized restart time, we upper bound its indicator pathwise by the sum of the corresponding events over all deterministic

restart indices $r$. Summing these probabilities and using the allocation $\sum_r \alpha_r \leq \alpha$ controls the expected contribution of such deviations.

**Step 3: Bounding the bias term.** Lemma A.3 controls the bias term $\mathcal{R}_T^{bias}$ by separating two contributions:
(i) *deviation events* where the empirical statistic underestimates its population counterpart. By Step 2, the sum of the probabilities of these events, over all relevant times and realized restart indices, is at most $\alpha$. Since each such event contributes at most $M^2$, their total contribution to the expected bias regret is at most $M^2\alpha$. (third term in Eq. (15)).
(ii) a *deterministic drift* term governed by the population statistic: If ATC detects the change, then $\max_k D_{k,t}^r$ crosses the threshold $\gamma_t^r$ within $O(\log(T/\alpha))$ steps, so the detection delay contributes only logarithmic bias regret. If ATC does not detect a change, we show that the resulting contribution to regret is at most logarithmic. Moreover, using the argument underlying Proposition 3.1, we show that the contamination carried into subsequent segments is likewise capped. As a result, the analysis can be repeated at the next change using the same detectable/undetectable dichotomy, yielding an overall bias term of order $\sigma^2 S \log(T/\alpha)$ (second term in Eq. (15)).

**Step 4: Bounding the variance term.** Lemma A.4 uses Step 2 to show that the expected number of false alarms is at most $\alpha$, and therefore the expected number of restarts $K$ is upper bounded by $\mathbb{E}[K] \leq S + 1 + \alpha$. Lemma A.5 shows that within a restart block of length $L$, summing the per-step variance over the block yields a contribution of order $\sigma^2(1 + \log L)$, where $L \leq T$. Summing over all $K$ blocks and using the bound on $\mathbb{E}[K]$ yields a bound of order $\sigma^2(S + 1 + \alpha)(1 + \log(T))$ (first term in Eq. (15)).

### 4.2. Lower bound

**Theorem 4.2** (*Regret lower bound*). *Assume $M \geq c_0\sigma$. There exists a universal constant $c > 0$ such that for all $T \geq 3$ and $1 \leq S < T$,*

$$
\mathcal{R}_T^\star(\mathcal{E}) \geq c \, \sigma^2 (S + 1) \left( 1 + \log\left( \left\lfloor \frac{T}{S+1} \right\rfloor + 1 \right) \right). \tag{16}
$$

The proof is given in App. A.5. The linear dependence on the number of changes $S$ and the logarithmic dependence on $T$ in Eq. (16) reflect an inherent cost of non-stationarity: each change induces an unavoidable logarithmic regret due to the time required to accumulate sufficient evidence for detection and to re-estimate the segment mean. The condition $M \geq c_0\sigma$ is a non-degeneracy condition ensuring that the model class contains mean shifts of order the noise scale. Th. 4.2 implies $\mathcal{R}_T^\star(\mathcal{E}) = \Omega\left(\sigma^2(S + 1)\log(T/(S+1))\right)$.

**Proof sketch.** We lower bound the minimax regret by isolating two unavoidable contributions that mirror the upper bound: (i) a bias cost incurred after changes, and (ii) an estimation-variance cost. To lower bound the bias from a single change of magnitude $\Delta$, we partition the horizon into $m$ windows of length $\ell$, inducing an $m$-ary hypothesis test over the change location. The window length must balance two effects: it should be large enough that missing the change incurs substantial regret, yet short enough to prevent sufficient information accumulation to distinguish a change from the no-change instance. Using a data-processing inequality, we show that any policy with controlled false alarms cannot reliably adapt to the change unless $\ell \gtrsim \frac{\sigma^2}{\Delta^2} \log T$. Since delayed adaptation incurs squared-loss regret of order $\Delta^2$ per step, this yields a lower bound of order $\Omega(\sigma^2 \log T)$. Repeating this argument over $S$ disjoint blocks of length $\Theta(T/S)$ gives a cumulative bias regret of order $\Omega(\sigma^2 S \log(T/S))$. For the variance term, we reduce to sequential mean estimation on each stationary segment. A Bayes–minimax argument yields a lower bound of $\Omega\!\left(\sigma^2 \sum_{j=0}^{S} \log(|\mathcal{I}_j| + 1)\right)$, which is maximized by an equal-length partition, giving $\Omega(\sigma^2 (S+1) \log(T/(S+1)))$.

### 4.3. Discussion

- Comparing Theorems 4.1 and 4.2 reveals a gap of order $\log(S)$ in the leading term between the upper and lower bounds. We conjecture that if the learner has prior knowledge of the horizon $T$ and the number of changes $S$, then a regret of order $O\!\left(\sigma^2 S \log(T/S)\right)$ is achievable, for example by restarting the estimator and test statistic after at most $T/(S+1)$ rounds. Whether such a bound is achievable without knowledge of $T$ and $S$ remains open.

- The lower bound admits a clean characterization in terms of the density of changes. Specifically, it scales as $\Omega\!\left(\sigma^2 \, S \log(T/S)\right)$, which is sublinear whenever the number of changes satisfies $S = o(T)$, and becomes linear in the regime $S = \Theta(T)$. This suggests that the information-theoretic cost of non-stationarity remains sublinear for any vanishing change density $S/T$.

- The logarithmic dependence on $T$ is tied to the squared-loss criterion, which is natural for mean estimation. For alternative losses, the rate can be different: for instance, under $L_1$ loss, one can prove a minimax lower bound of $\Omega(\sqrt{ST})$ (details omitted). Establishing regret guarantees under different loss functions is left for future work.

- In $\mathbb{R}^d$, the regret decomposition remains the same, but the concentration bound introduces an additional $\sqrt{d}$ term in the threshold and the within-block estimation variance scales as $d\sigma^2/(t-r)$. Consequently, the variance part scales as $O\!\left(d\sigma^2 (S+1) \log(T)\right)$ and the bias part gains an additive $O(\sigma^2 S d)$ term, yielding an overall dynamic re-

gret bound that matches the scalar logarithmic dependence on $T$ up to natural $d$-dependent factors. More details are given in App. A.7.

## 5. Numerical results

We evaluate the proposed ATC algorithm on both synthetic and real-world data. Full implementation details, along with additional comprehensive experiments, are provided in App. A.1.[3]

**Synthetic experiments**. Our primary goal is to empirically validate the theoretical regret scaling in Th. 4.1 and to examine ATC's behavior under abrupt, unstructured mean shifts. We consider a piecewise-stationary environment with a fixed number of $S = 5$ changes and vary the horizon $T$. Figure 3 summarizes the results. Fig. 3(a) shows a representative realization with $T = 1200$, and Fig. 3(b) shows the cumulative regret on this environment. Each change induces a transient regret increase due to bias from aggregating samples across regimes. When a change is missed (e.g., the second and third shifts), the estimator operates on mixed-regime data, inducing endogenous confounding and consequently increasing the regret in that segment. Despite missed detections, the information loss remains controlled and does not prevent subsequent detection (near $t \approx 900$), consistent with the theory. Fig. 3(c) reports cumulative regret averaged over 5000 Monte Carlo runs as a function of $\log T$ for ATC and its computationally efficient variant (Section 3.3). In both cases, regret grows linearly with $\log T$, confirming the theoretical scaling. The curves differ by a constant offset, indicating that the approximation preserves asymptotic behavior up to constant factors.

Due to space limitations, additional simulation results are deferred to App. A.1. These experiments examine: (i) dense-change regimes, characterizing the change-point densities for which sublinear regret is no longer observed, in line with the theory (App. A.1.2); (ii) long-horizon behavior, highlighting the anytime nature of ATC and showing that constant thresholds may be preferable in some finite known-horizon regimes, whereas the ATC threshold becomes advantageous when the horizon is long or unknown (App. A.1.4); (iii) sensitivity to variance misspecification (App. A.1.5); and (iv) adversarial environments, illustrating the segment lengths and jump magnitudes that lead to large regret (App. A.1.6).

**Real-world data: NAB benchmark.** To complement the synthetic experiments, we evaluate ATC on a benchmark derived from the Numenta Anomaly Benchmark (NAB) dataset (Lavin & Ahmad, 2015), which records CPU usage from Amazon Web Services (AWS) by measuring average

---

[3]The implementation used in this paper is publicly available at https://github.com/gafnito/cost-of-learning-multiple-change-points.

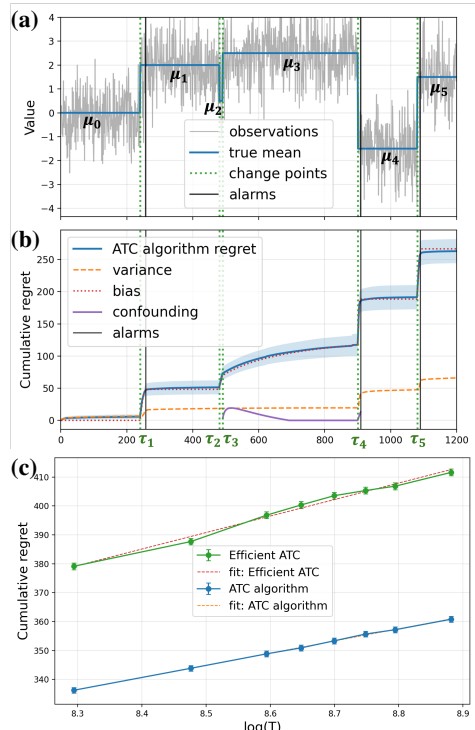

*Figure 3.* Synthetic environment and regret scaling for ATC.

## 6. Conclusion and Future Work

We present a learning-theoretic formulation for multiple change-point environments, which gives rise to an under-explored phenomenon: unavoidable missed detections induce endogenous confounding, in which the reference distribution used for detection becomes a mixture of past regimes. We formulate this through a regret-minimization framework for online tracking, without imposing lower bounds on separability or spacing between changes. We quantify the cost of confounding via a reduction in the detector SNR, which requires a novel analysis exploiting the mixture structure of the learner's statistic and motivates a time-varying detection threshold that is robust to missed detections. Leveraging these insights, our proposed ATC algorithm achieves nearly minimax-optimal regret without knowledge of the horizon or the number of changes.

There are several natural extensions of this work. The mean-tracking model was chosen as a canonical setting that isolates the statistical difficulty imposed by multiple change points themselves; nevertheless, the same perspective may extend to broader observation models. For example, in exponential-family models, an ATC-type procedure could track the expectation parameter through empirical sufficient statistics, with the analysis depending on concentration and local curvature properties of the model. Similar ideas may also extend to regression or dynamical settings with time-varying parameters and temporally dependent observations. Another important direction is to relax the assumption that the variance proxy is known, by incorporating online variance calibration into both the estimator and the detection threshold. Finally, gradually changing environments raise a related but distinct bias-variance tradeoff: past samples remain useful while the accumulated drift over a window is below the statistical estimation scale, but become harmful once this drift dominates the variance reduction from pooling. Developing regret guarantees for these extensions, as well as for broader decision-making settings, is an interesting direction for future work.

utilization across a cluster (Fig. 4(a)). Non-stationarity may arise from software upgrades and configuration changes that occur at arbitrary times and alter system behavior. Accurate tracking is operationally important, since sustained high utilization triggers machine allocation, while low utilization leads to de-allocation. We compare ATC against sliding-window and discounted-mean baselines, with parameters tuned offline (see App. A.1). Fig. 4(b) shows that ATC achieves the lowest cumulative regret by explicitly detecting change points and restarting, thereby preventing old-regime samples from biasing the estimate. In contrast, the baselines adapt more slowly and incur large regret spikes after shifts, most notably following the large jump near the end. Overall, explicit resets enable faster adaptation and improved robustness to abrupt regime changes.

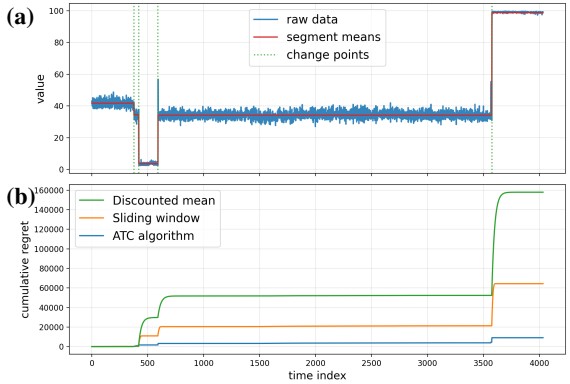

*Figure 4.* Cumulative regret on the NAB CPU dataset.

## Impact Statement

This paper presents work whose goal is to advance the field of Machine Learning. There are many potential societal consequences of our work, none which we feel must be specifically highlighted here.

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

# A. Appendix

## A.1. Extended Simulations

We present additional simulation results that further illustrate the behavior of ATC under challenging non-stationary environments and clarify the tradeoffs underlying its design. We first describe the full details to reproduce the simulations presented in Section 5.

### A.1.1. Implementation details for main simulations.

**Synthetic experiments.** We describe here the components required to reproduce the experiments in Fig. 3. The observation sequence satisfies $X_t = \mu_t + \sigma Z_t$ with $\sigma = 1$ and $Z_t \sim \mathcal{N}(0,1)$ i.i.d. The mean process has $S = 5$ changes placed at fixed fractions of the horizon: $\tau_1 = \lfloor 0.2T \rfloor + 1$, $\tau_2 = \lfloor 0.4T \rfloor + 1$, $\tau_3 = \tau_2 + 10$, $\tau_4 = \lfloor 0.75T \rfloor + 1$, and $\tau_5 = \lfloor 0.9T \rfloor + 1$, with $\tau_0 = 1$ and $\tau_6 = T + 1$. The third segment has fixed length 10, creating a short-lived regime designed to induce missed detections and illustrate endogenous confounding. The corresponding segment means are $(0, 2, 0.5, 2.5, -1.5, 1.5)$, as visualized in Fig. 3(a). We use $\alpha = 0.05$ throughout.

Figure 3(b) shows the cumulative squared-error $\sum_{s=1}^{t}(\hat{\mu}_s - \mu_s)^2$ for a single realization with $T = 1200$. For Fig. 3(c), we estimate the expected regret $\mathcal{R}_T = \sum_{t=1}^{T}(\hat{\mu}_t - \mu_t)^2$ by averaging over 5000 independent Monte Carlo replications for each $T \in \{600, 1200, 2400, 4800, 7000, 9000\}$, and we report $95\%$ confidence intervals ($\pm 1.96$ standard errors). For the efficient variant of ATC we set $b = 2$ (Section 3.3).

**NAB benchmark.** We use the NAB time series *ec2_cpu_utilization_ac20cd.csv* from the NAB dataset (Lavin & Ahmad, 2015), which includes AWS server metrics as collected by the Amazon CloudWatch service, and extract the CPU utilization values from the second CSV column, yielding a sequence $X_1, \ldots, X_T$ of length $T$. To define a reproducible reference target for regret evaluation, we convert a fixed, manually specified list of change-point indices $\mathcal{T} = \{377, 420, 592, 3575\}$ into a piecewise-constant mean sequence $\mu_t$ by segmenting the series at $\tau_0 = 1$, $\tau_1, \ldots, \tau_S \in \mathcal{T}$, and $\tau_{S+1} = T + 1$, and setting

$$\mu_t = \frac{1}{\tau_{j+1} - \tau_j} \sum_{s=\tau_j}^{\tau_{j+1}-1} X_s, \qquad \text{for } t \in [\tau_j, \tau_{j+1}).$$

For a fair comparison, baseline parameters and $\sigma$ were tuned offline on a separate collection of environments and then fixed when evaluating on the specific NAB time series shown in the figure. The sliding-window baseline uses a fixed window length $W = 30$:

$$\hat{\mu}_t^{\text{SW}} = \frac{1}{\min\{W, t\}} \sum_{i=\max\{1, t-W+1\}}^{t-1} X_i,$$

and the discounted-mean baseline uses an exponentially weighted average with discount factor $\rho = 0.98$:

$$\hat{\mu}_t^{\text{DM}} = \frac{\sum_{i=1}^{t-1} \rho^{t-i} X_i}{\sum_{i=1}^{t-1} \rho^{t-i}}.$$

For ATC we use the same thresholding rule as in the synthetic experiments, with user-level $\alpha = 0.05$ and $\sigma = 1$. Regret traces in Panel 4(b) are computed for each algorithm from its estimate $\hat{\mu}_t$ and the fixed reference $\mu_t$ above.

### A.1.2. Regret under dense change points.

We examine how the density of changes affects regret by allowing the number of change points $S$ to grow with the horizon $T$. Theorem 4.1 predicts a transition around $S = \Theta(T/\log T)$. When $S$ is of this order, the upper bound permits regret linear in $T$, whereas if $S = o(T/\log T)$ the regret is expected to remain sublinear (see Section 4.3). Figure 5a illustrates this transition by comparing a dense regime $S(T) \asymp T/\log T$ to a slightly sparser scaling $S(T) = T^{0.95}/\log T$. Consistent with the theory, the dense regime exhibits near-linear growth in $T$, whereas the sparser regime remains sublinear over the simulated horizons.

**Implementation details.** Both scenarios in Figure 5a use the same observation model as in the synthetic experiments, $X_t = \mu_t + \sigma Z_t$ with $Z_t \sim \mathcal{N}(0,1)$ i.i.d. and $\sigma = 1$, but the number of change points $S$ grows with the horizon $T$ and the

jump size is calibrated to the (shrinking) segment length, where

$$T \in \{1200, 2400, 4800, 7000, 9000, 12000, 15000, 18000, 20000, 22000, 25000\}.$$

For each $T$, change points are placed evenly by partitioning $\{1, \ldots, T\}$ into $S+1$ consecutive segments of lengths differing by at most 1 (i.e., $\tau_0 = 1$, $\tau_{S+1} = T + 1$, and $\tau_{j+1} - \tau_j \in \{\lfloor a \rfloor, \lceil a \rceil\}$ where $a = T/(S+1)$). The mean alternates between two levels with amplitude $\Delta(T)$:

$$\mu_j \in \{0, \Delta(T)\}, \qquad \mu_j = \begin{cases} 0, & j \text{ even}, \\ \Delta(T), & j \text{ odd}, \end{cases} \qquad j = 0, 1, \ldots, S,$$

so every change has magnitude $\Delta(T)$. The dense regime uses

$$S(T) = \left\lfloor \frac{T}{\log T} \right\rfloor,$$

whereas in the sparser regime we set

$$S(T) = \left\lfloor \frac{T^\delta}{\log T} \right\rfloor, \qquad \delta = 0.95.$$

In both cases the jump amplitude is calibrated as

$$\Delta(T) \;=\; \sqrt{\frac{160\,\sigma^2 \log a}{a}}, \qquad a = \frac{T}{S(T) + 1},$$

which keeps the per-segment signal-to-noise ratio at the detection boundary as segment lengths shrink. For each $T$, we run 1000 independent Monte Carlo replications, and evaluate ATC with $\alpha = 0.05$ and $\sigma = 1$. Figure 5a plots the Monte Carlo mean of $\mathcal{R}_T$ against $T$ with 95% confidence intervals ($\pm 1.96$ standard errors).

### A.1.3. COMPARISON WITH PASSIVE ALGORITHMS.

We further compare ATC in the same synthetic environment as in Section 5 to the passive baselines as in App. A.1.1. Fig. 5b shows that ATC achieves the lowest cumulative regret by explicitly detecting change points and restarting, whereas the baselines adapt more slowly. Overall, explicit resets enable faster adaptation and improved robustness to abrupt regime changes.

**Implementation details.** All experiments use the same synthetic environment as in Section 5, with a fixed number of $S = 5$ change points and horizon values

$$T \in \{600, 1200, 2400, 3000, 3800, 4800, 5500, 6300, 7000, 8000, 9000\}.$$

We report cumulative squared-error regret averaged over 1000 Monte Carlo replications.

### A.1.4. TIME-VARYING VERSUS CONSTANT THRESHOLDS.

We now isolate the role of the time-varying threshold in the anytime setting by comparing ATC with several constant-threshold detectors. Specifically, we consider three fixed thresholds of the form $\gamma = c\sigma$, including the aggressive choice $\gamma = 4.81\sigma$ suggested by finite-horizon tuning, as well as more conservative alternatives. Figure 6 summarizes the comparison in terms of average regret (left) and false-alarm (FA) rates (right) as a function of the horizon $T$.

As shown in Figure 6a, the threshold $\gamma = 4.81\sigma$ achieves the lowest regret for small and moderate horizons. This behavior is explained by its high detection rate: early in the horizon, rapid alarms lead to short detection delays and reduced bias after each change. However, this improved adaptivity comes at the cost of an increasing false-alarm rate, shown in Figure 6b. In particular, the FA rate of $\gamma = 4.81\sigma$ grows steadily with $T$. In contrast, ATC maintains a near-zero FA rate uniformly over the horizon by increasing its threshold over time, illustrating its anytime guarantee.

As $T$ increases, the accumulation of false alarms under $\gamma = 4.81\sigma$ leads to excessive restarts and therefore to information loss (since discarding samples increases the estimator variance), which ultimately degrades performance and causes its regret to surpass that of ATC. In contrast, ATC adaptively raises its threshold over time, preserving a controlled false-alarm rate while still detecting sufficiently strong changes.

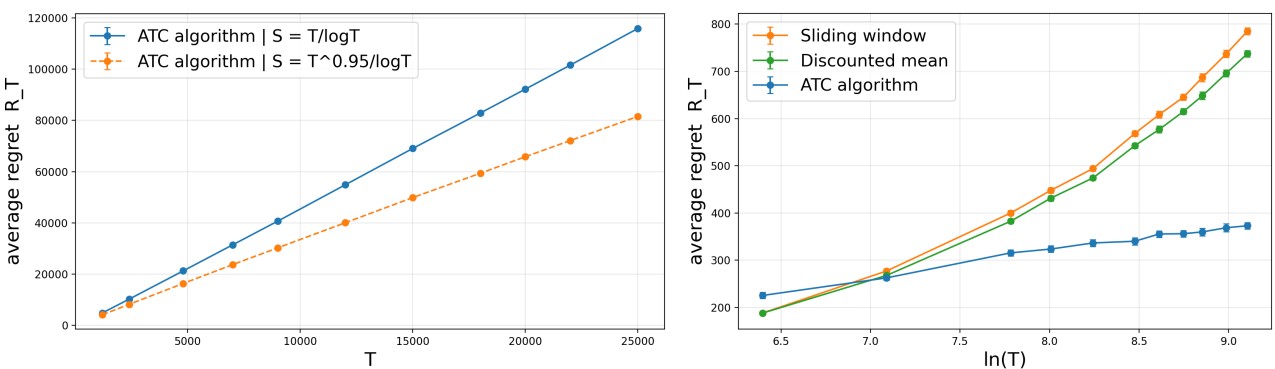

(a) Cumulative regret as a function of the horizon $T$ in a dense change-point regime.

(b) Cumulative regret comparison between ATC and passive baseline algorithms.

*Figure 5.* Dense change points and passive algorithms.

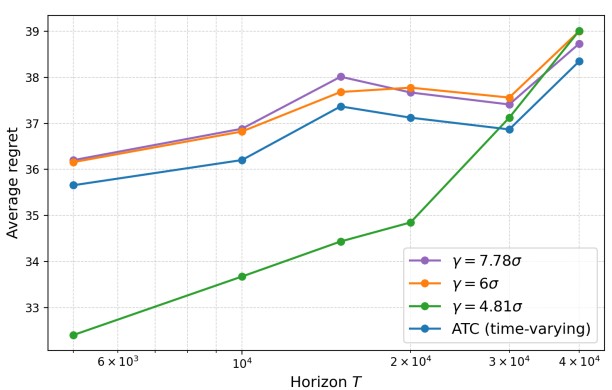

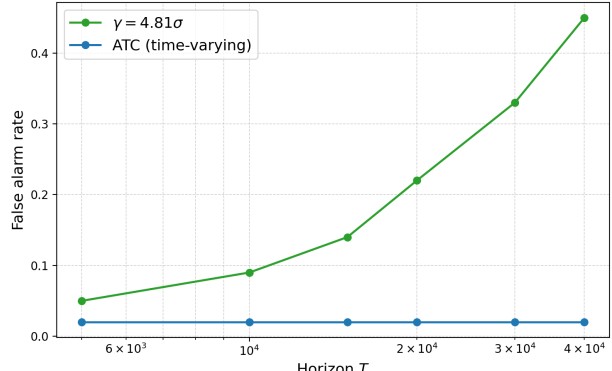

(a) Average regret as a function of the horizon $T$ for ATC and constant thresholds.

(b) False-alarm (FA) rate as a function of the horizon $T$. Only ATC and $\gamma = 4.81\sigma$ are shown, as the remaining constant thresholds coincide with ATC.

*Figure 6.* Comparison between ATC and constant-threshold detectors in the anytime setting.

**Implementation details.** Figure 6 is generated using a single-change synthetic environment designed to emphasize early–time behavior. For each run we simulate observations $X_t = \mu_t + \sigma Z_t$ with $Z_t \sim \mathcal{N}(0,1)$ i.i.d. and $\sigma = 1$, where the mean is piecewise-constant with a single change point at

$$\tau_1 = 50, \qquad \mu_t = \begin{cases} 0, & 1 \le t < \tau_1, \\ 0.75, & \tau_1 \le t \le T, \end{cases}$$

so $S = 1$ and the jump magnitude is $0.75$, and

$$T \in \{5000, 10000, 15000, 20000, 30000\}.$$

We average all reported quantities over 1000 Monte Carlo replications.

### A.1.5. REAL-WORLD DATA: VARIANCE-PROXY MISSPECIFICATION.

We examine the sensitivity of ATC to misspecification of the sub-Gaussian noise parameter $\sigma$. Since the noise variance is not known a priori for real-world data, we evaluate ATC under several fixed choices of $\sigma$, including $\sigma = 1$, which is estimated offline from the data and yields the best empirical performance. As shown in Figure 7b, when $\sigma \in \{1, 4\}$, ATC reliably tracks the evolving mean and incurs only transient bias following each change. In contrast, under the underestimated choice $\sigma = 0.5$, the algorithm triggers an excessive number of false alarms, leading to frequent restarts and diverging cumulative

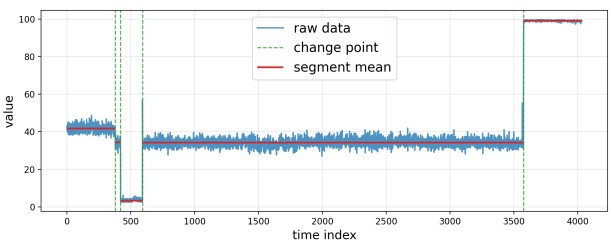

*(a)* CPU usage time series with change points.

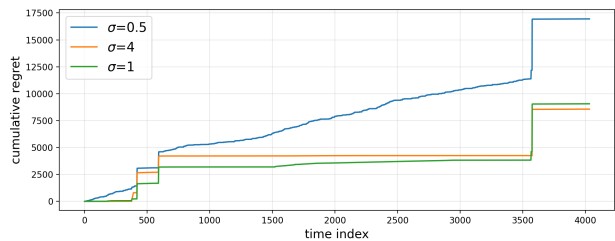

*(b)* Cumulative regret of ATC for different noise variance choices.

*Figure 7.* ATC on real-world data from the NAB benchmark (AWS CPU usage).

regret. This highlights the sensitivity of change detection to variance misspecification in real data. Developing adaptive or robust mechanisms to mitigate the impact of variance misspecification is an interesting direction for future work.

### A.1.6. ADVERSARIAL ENVIRONMENT.

We construct a deliberately "hard" (adversarial) environment for ATC by placing change points at equal spacing $a \approx T/(S+1)$ and alternating the mean across segments by a gap $\Delta$. This construction exposes a fundamental tradeoff in regret. On the one hand, smaller values of $\Delta$ reduce the instantaneous loss incurred after a change, but lead to longer detection delays. On the other hand, larger values of $\Delta$ are detected more rapidly, but induce larger losses before the restart.

A simple expectation-level calculation highlights the most challenging regime. Let $a$ denote the number of pre-change samples and $b_t$ the number of post-change samples observed since the shift. Then the cumulative regret accumulated up to time $t$ after the change satisfies

$$R(t) \leq \min\left\{a\Delta^2, \ \sigma^2\big(\gamma_t^r\big)^2\right\},$$

where the first term reflects the intrinsic ceiling $R(t) \leq a\Delta^2$, while the second term arises from the drift–threshold comparison: before an alarm is triggered, the effective information $\frac{ab_t}{a+b_t}\frac{\Delta^2}{\sigma^2}$ cannot exceed $(\gamma_t^r)^2$. This deterministic upper bound is maximized near the boundary $a\Delta^2 \asymp \sigma^2(\gamma_t^r)^2$, suggesting that the most adverse shifts are those calibrated close to this regime. In our simulations, we instantiate this stress test by setting

$$\Delta^2 \ = \ c\,\frac{\sigma^2 \log a}{a}.$$

The constant $c$ controls proximity to the boundary: smaller values push the shifts toward missed detection, whereas larger values make detection easier but increase the instantaneous loss.

The resulting environment is illustrated in Figure 8a. Figure 8b compares the performance of ATC on this adversarial construction with the milder environment shown in Figure 3. As predicted by the theoretical analysis, both environments exhibit logarithmic regret growth. However, the adversarial instance incurs noticeably higher regret, since each change is deliberately placed near the deterministic detection boundary, where the bound $\min\{a\Delta^2, \sigma^2(\gamma_t^r)^2\}$ is largest. In this regime, detection is delayed long enough for bias to accumulate over a non-negligible fraction of each segment, leading to sustained information dilution prior to each restart. As a result, each change contributes close to its maximal admissible regret, and these contributions aggregate across the $S$ changes. By contrast, in the milder environment, many changes are detected earlier and therefore contribute substantially less to the cumulative regret.

**Implementation details.** The adversarial experiment fixes the number of change points to $S = 5$ and places them evenly along the horizon, with segment length $a \approx T/(S+1)$. For each

$$T \in \{600, 1200, 2400, 4800, 7000, 9000\},$$

we construct a piecewise-constant mean sequence with alternating levels $\mu_j \in \{0, \Delta\}$ across segments with $\Delta$ chosen as described above. The constant $c$ controls the difficulty of detection: the non-adversarial experiment corresponds to $c = 1000$, whereas the adversarial setting uses $c = 160$, which optimizes the tradeoff described above. For each $(T, c)$ pair we generate 1000 independent Monte Carlo replications.

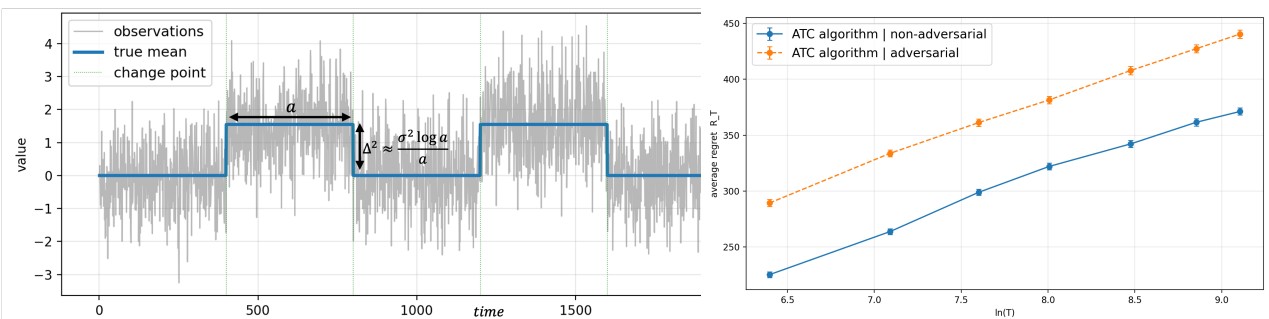

*(a)* Adversarial environment, where the magnitude of the shift is chosen to maximize the regret.

*(b)* Comparison between the cumulative regret of ATC under adversarial and non-adversarial environments.

*Figure 8.* Adversarial environment.

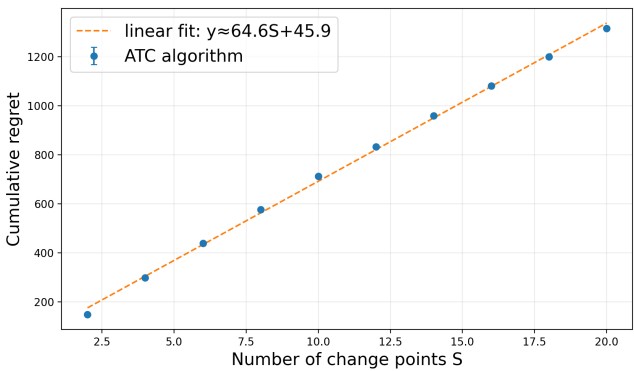

*Figure 9.* Empirical scaling of cumulative regret with the number of change points $S$.

All runs use ATC with the time-varying threshold $\gamma_t^r$ and confidence parameter $\alpha = 0.05$. Figure 8b reports the Monte Carlo average of $\mathcal{R}_T$ over 1000 runs of as a function of $T$ for the two choices of $c$.

A.1.7. REGRET SCALING IN THE NUMBER OF CHANGES

Finally, we empirically validate the linear dependence of the regret on the number of change points $S$, as established in Theorem 4.1. As shown in Figure 9, the cumulative regret scales linearly with $S$, matching the predicted order and supporting the tightness of the bound. We use the same observation model as in the previous experiments, with $\sigma = 1$ and $\alpha = 0.05$. The number of change points is varied from 2 to 20 in increments of 2, while the horizon is fixed at $T = 1000$. All results are averaged over 1000 Monte Carlo runs.

**A.2. Notations**

Recall that $N_G^r$ in Eq. (6) denotes the alarm time (i.e., stopping time) of the ATC test when started at time $r$. After each alarm, the algorithm restarts both the threshold and the estimator by setting $r \leftarrow t - 1$. We denote the total number of alarms to be $K$. Starting from $r_0 := 1$, we define the (random) sequence of restarts recursively by

$$r_{m+1} := (N_G^{r_m} - 1) \wedge T, \quad m = 0, 1, \ldots, \tag{17}$$

and stop at the first $K$ such that $r_K = T$. For each restart $r_m$ with $m = 0, \ldots, K-1$, define the corresponding block

$$B_m := \{t \in \{2, \ldots, T\} : r_m < t \leq r_{m+1}\}. \tag{18}$$

By construction, the random sets $\{B_m\}_{m=0}^{K-1}$ are disjoint and form a partition of $\{2, \ldots, T\}$: for every $t \in \{2, \ldots, T\}$ there is a unique $m$ such that $t \in B_m$. Let $r(j)$ denote the last alarm before $\tau_j$, i.e.,

$$r(j) = \max\{r_m : r_m < \tau_j\}. \tag{19}$$

Figure 10 illustrates the notation used in the construction of the upper bound proof.

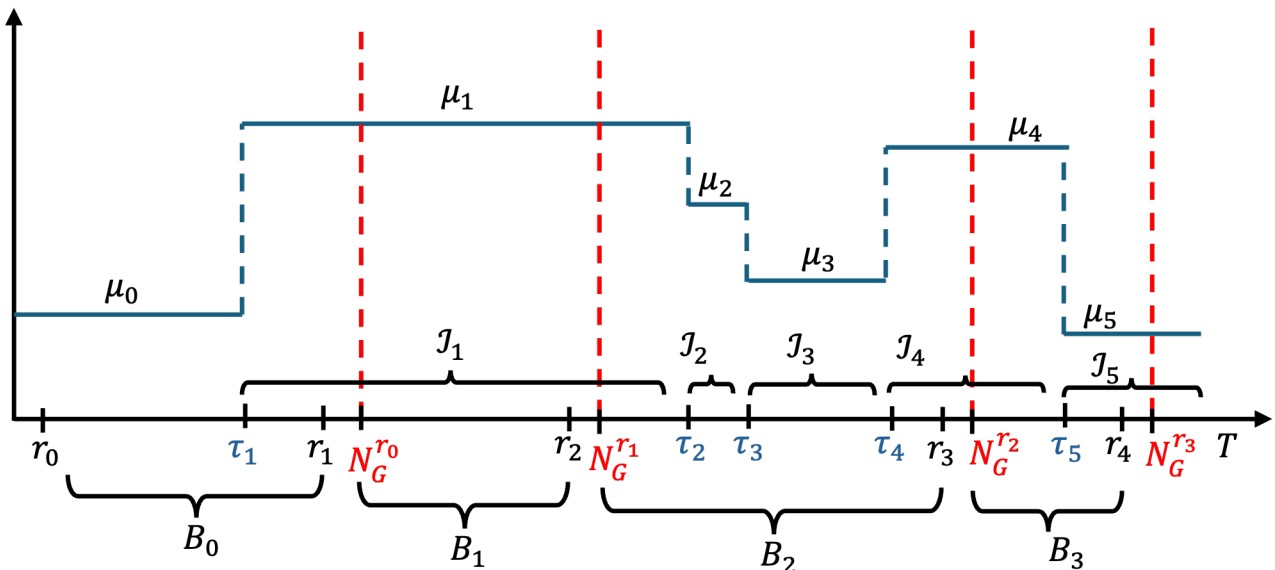

*Figure 10.* Schematic Illustration of the block and restart structure used in the upper-bound analysis. The blue solid line shows the underlying piecewise-stationary mean $\mu_t$ with true change points $\{\tau_j\}$. Red dashed lines denote alarm times $\{N_G^r\}$ at which the algorithm restarts, inducing blocks $B_m$. A block may contain several true change points. $\mathcal{I}_j = [\tau_j, \tau_{j+1})$ as defined in Section 2.

### A.3. Proof of Theorem 4.1

In this section we prove the regret upper bound for ATC using the notations introduced in App. A.2. The argument proceeds in five steps. First, we decompose the regret into a global variance term and a bias term, aligned with the block structure induced by the alarms of the algorithm. Second, we prove a fixed-restart anytime deviation inequality for the GLR statistic $\widehat{D}_{k,t}^r$ in Eq. (4). This inequality is used through its summable probability budget: in the regret analysis, events evaluated at random restart times are bounded pathwise by sums over deterministic restart indices, and their probabilities are then summed. This controls the contribution of rare deviation events. We use this result to bound the bias regret and the expected number of false alarms, which is then leveraged to bound the variance regret. When there is no ambiguity, we write $\mathcal{R}_T \equiv \mathcal{R}_T^{\text{ATC}}$. Since the estimate at the first time step is unspecified, we evaluate the regret starting from $t = 2$.

**Lemma A.1** (*Regret upper bound decomposition*). *The regret can be decomposed as*

$$\mathcal{R}_T \leq 2\,\mathbb{E}\left[\sum_{m=0}^{K-1} \sum_{t\in B_m} \left(\hat{\mu}_t^{r_m} - \bar{\mu}_t^{r_m}\right)^2\right]$$
$$+ 2\,\mathbb{E}\left[\sum_{j=1}^{S} \left(\Delta_j^{(\text{eff})}\right)^2 \sum_{t=\tau_j}^{\tau_{j+1}-1} \left(\frac{a_L^{(j)}}{a_L^{(j)} + b_t}\right)^2 \mathbf{1}\{N_G^{r(j)} > t\}\right]$$
$$=: \mathcal{R}_T^{\text{var}} + \mathcal{R}_T^{\text{bias}}, \tag{20}$$

*with $K$ and $r(j)$ defined in App. A.2, $\bar{\mu}_t^{r_m}$ defined in Eq. (26), $a_L^{(j)}$ and $b_t$ defined in Eq. (29) and $\Delta_j^{(\text{eff})}$ defined in Eq. (31).*

The proof is given in App. A.4.1. The term $\mathcal{R}_T^{\text{var}}$ is a global variance-like term that collects all noise contributions around the block means. The term $\mathcal{R}_T^{\text{bias}}$ captures the error due to distribution shifts: it is nonzero only on blocks containing true change points and only before the algorithm has raised an alarm and restarted the estimator.

The next lemma provides a uniform high-probability bound on the deviation between the empirical statistic $\widehat{D}_{k,t}^r$ and its population version $D_{k,t}^r$ defined in Eq. (9).

**Lemma A.2** (*High-probability confidence bound*). *For each $r, t \in [T]$ with $t > r + 1$, let $\gamma_t^r$ be defined as in Eq. (7). Then, for each fixed $r$ we have*

$$\mathbb{P}\left(\exists\, k, t \in [T],\ r < k < t : \left|\hat{D}_{k,t}^r - D_{k,t}^r\right| \geq \gamma_t^r\right) \leq \alpha_r. \tag{21}$$

The proof is given in App. A.4.2. Lemma A.2 provides an upper bound of $\alpha_r$ on the deviation probability for each deterministic restart index $r$, uniformly over all split points $k$ and times $t$. In the regret analysis below, we use this result through the resulting summable probability budget. In particular, when a bad event is evaluated at a random restart time, we dominate its indicator by the sum of the same events over all deterministic restart indices and then sum the corresponding probabilities. This avoids conditioning on a random restart time and yields expectation-level control of the stochastic bias term and of the number of false alarms.

**Lemma A.3** (*Bias regret upper bound*). *Under the bounded mean diameter $\Delta_{diam} \leq M$ (Eq. (1)), we have*

$$\mathcal{R}_T^{\text{bias}} \leq 2M^2\alpha + 2M^2 S + C_B\,\sigma^2\,S \log\left(\frac{T}{\alpha}\right), \tag{22}$$

*for some universal constant $C_B > 0$.*

The proof is given in App. A.4.3. The first term in Eq. (22) corresponds to the contribution of rare deviation events where the empirical statistic substantially underestimates its population version; it is controlled by the confidence level $\alpha$. The second and third terms capture the accumulated regret before ATC detects a change and restarts its estimator: each change point contributes at most a logarithmic factor $\sigma^2 \log(T/\alpha)$ in bias regret, leading to the overall $O(\sigma^2 S \log(T/\alpha))$ dependence.

We next show, using Lemma A.2, that the expected number of false alarms is upper bounded by $\alpha$.

**Lemma A.4** (*Expected number of blocks*). *Let $N_{\text{FA}}$ denote the total number of false alarms produced by the ATC algorithm up to time $T$, and let $K$ be the total number of blocks (restarts). Then*

$$\mathbb{E}[N_{\text{FA}}] \leq \sum_{r=1}^{\infty} \alpha_r \leq \alpha,$$

*and hence*

$$\mathbb{E}[K] \leq S + 1 + \alpha.$$

The proof is given in App. A.4.4. Lemma A.4 will be used to bound the variance term in Eq. (20).

**Lemma A.5** (*Variance regret upper bound*). *There exists an absolute constant $C_V > 0$ such that*

$$\mathcal{R}_T^{\text{var}} \leq C_V\,\sigma^2\,(S + 1 + \alpha)\,(1 + \log T). \tag{23}$$

The proof is given in App. A.4.5. Here, within each block between two alarms, the running average noise can be controlled by a harmonic-series argument, so its variance at age $s$ is of order $\sigma^2/s$, with the selected restart sample handled separately. Summing over a block of length $L$ yields a harmonic series of order $1 + \log L$, and summing over all blocks leads to the stated bound once we control the expected number of blocks by $S + 1 + \alpha$.

Combining Lemma A.3 and Lemma A.5, we obtain the claimed regret bound in Theorem 4.1.

## A.4. Proofs of Lemmas

A.4.1. PROOF OF LEMMA A.1

*Proof.* We first write the regret as the sum over the disjoint blocks induced by the algorithm

$$\mathcal{R}_T = \mathbb{E}\left[\sum_{m=0}^{K-1} \sum_{t \in B_m} \left(\hat{\mu}_t^{r_m} - \mu_t\right)^2\right]. \tag{24}$$

Next, for any $t \in B_m$ we decompose the squared error as

$$
\begin{aligned}
\left(\hat{\mu}_t^{r_m} - \mu_t\right)^2 &= \left[\left(\hat{\mu}_t^{r_m} - \bar{\mu}_t^{r_m}\right) + \left(\bar{\mu}_t^{r_m} - \mu_t\right)\right]^2 \\
&\leq 2\left(\hat{\mu}_t^{r_m} - \bar{\mu}_t^{r_m}\right)^2 + 2\left(\bar{\mu}_t^{r_m} - \mu_t\right)^2,
\end{aligned}
\tag{25}
$$

with

$$
\bar{\mu}_t^{r_m} := \frac{1}{t - r_m} \sum_{i=r_m}^{t-1} \mu_i.
\tag{26}
$$

Summing over all blocks and all times and taking expectations gives

$$
\begin{aligned}
\mathcal{R}_T &= \mathbb{E}\left[\sum_{m=0}^{K-1} \sum_{t \in B_m} \left(\hat{\mu}_t^{r_m} - \mu_t\right)^2\right] \\
&\leq 2\,\mathbb{E}\left[\sum_{m=0}^{K-1} \sum_{t \in B_m} \left(\hat{\mu}_t^{r_m} - \bar{\mu}_t^{r_m}\right)^2\right] + 2\,\mathbb{E}\left[\sum_{m=0}^{K-1} \sum_{t \in B_m} \left(\bar{\mu}_t^{r_m} - \mu_t\right)^2\right].
\end{aligned}
\tag{27}
$$

The first term in Eq. (27) collects all variance-like contributions around the block means $\bar{\mu}_t^{r_m}$. The second term is a pure bias term, and it is nonzero only at times $t$ for which the block mean $\bar{\mu}_t^{r_m}$ differs from the segment mean $\mu_t$, i.e., only after a true change point within the block.

Recall that $r(j)$ denotes the last alarm before $\tau_j$. For any $t \in [\tau_j, \tau_{j+1})$, if an alarm occurs after $\tau_j$, the subsequent restart time satisfies $N_G^{r(j)} - 1 \geq \tau_j$, hence $\bar{\mu}_t^{N_G^{r(j)}-1} = \mu_j$ and the bias term vanishes; consequently, the bias on segment $j$ is nonzero only on $\{N_G^{r(j)} > t\}$.

Since before the first change point the bias term is zero, we can write the second term in Eq. (27) as:

$$
2\,\mathbb{E}\left[\sum_{m=0}^{K-1} \sum_{t \in B_m} \left(\bar{\mu}_t^{r_m} - \mu_t\right)^2\right] = 2\,\mathbb{E}\left[\sum_{j=1}^{S} \sum_{t=\tau_j}^{\tau_{j+1}-1} \left(\bar{\mu}_t^{r(j)} - \mu_t\right)^2 \cdot \mathbf{1}\{N_G^{r(j)} > t\}\right].
\tag{28}
$$

On the interval $\mathcal{I}_j$, define the *pre-change effective size* $a_L^{(j)}$ and the *post-change size* $b_t$ to be

$$
a_L^{(j)} := \tau_j - r(j), \qquad b_t := t - \tau_j.
\tag{29}
$$

Define the *effective pre-change mean* to be

$$
\mu_L^{(j)} := \frac{1}{a_L^{(j)}} \sum_{i=r(j)}^{\tau_j - 1} \mu_i,
\tag{30}
$$

so that $\mu_L^{(j)}$ is the average of the true means over the left (pre-change with respect to $\tau_j$) part of the block, i.e., indices $i \in \{r(j), \ldots, \tau_j - 1\}$. Finally, define the *effective mean gap*

$$
\Delta_j^{(\text{eff})} := \left|\mu_L^{(j)} - \mu_j\right|.
\tag{31}
$$

Therefore, for $t \in \mathcal{I}_j$ we have

$$
\bar{\mu}_t^{r(j)} = \frac{1}{a_L^{(j)} + b_t} \left(\sum_{i=r(j)}^{\tau_j - 1} \mu_i + \sum_{i=\tau_j}^{t-1} \mu_j\right) = \frac{a_L^{(j)} \mu_L^{(j)} + b_t\, \mu_j}{a_L^{(j)} + b_t},
$$

and hence

$$
\bar{\mu}_t^{r(j)} - \mu_j = \frac{a_L^{(j)}}{a_L^{(j)} + b_t} \left(\mu_L^{(j)} - \mu_j\right),
\tag{32}
$$

so that

$$\left(\bar{\mu}_t^{r(j)} - \mu_j\right)^2 = \left(\frac{a_L^{(j)}}{a_L^{(j)} + b_t}\right)^2 \left(\Delta_j^{(\text{eff})}\right)^2, \qquad t \in \mathcal{I}_j. \tag{33}$$

Summing over all segments yields

$$2\,\mathbb{E}\left[\sum_{j=1}^{S}\sum_{t=\tau_j}^{\tau_{j+1}-1}\left(\bar{\mu}_t^{r(j)} - \mu_t\right)^2 \cdot \mathbf{1}\{N_G^{r(j)} > t\}\right] = 2\,\mathbb{E}\left[\sum_{j=1}^{S}\sum_{t=\tau_j}^{\tau_{j+1}-1}\left(\frac{a_L^{(j)}}{a_L^{(j)} + b_t}\right)^2 \left(\Delta_j^{(\text{eff})}\right)^2 \cdot \mathbf{1}\{N_G^{r(j)} > t\}\right]. \tag{34}$$

$\square$

### A.4.2. PROOF OF LEMMA A.2

*Proof.* Fix $r \geq 1$ and $\alpha_r \in (0,1)$. For $r < k < t$ define

$$A_{k,t}^r := \frac{1}{\sigma}\left(\frac{t-k}{(k-r)(t-r)}\right)^{1/2}\sum_{i=r}^{k-1}(X_i - \mu_i) - \frac{1}{\sigma}\left(\frac{k-r}{(t-k)(t-r)}\right)^{1/2}\sum_{i=k}^{t-1}(X_i - \mu_i). \tag{35}$$

By construction, $\mathbb{E}[A_{k,t}^r] = 0$. Define $Z_i = X_i - \mu_i$. By the sub-Gaussian assumption, for every $i$ and every $\lambda \in \mathbb{R}$,

$$\mathbb{E}\left[e^{\lambda Z_i}\right] \leq \exp\left(\frac{\sigma^2\lambda^2}{2}\right).$$

Write $A_{k,t}^r = \sum_{i=r}^{t-1} a_i Z_i$ with

$$a_i := \frac{1}{\sigma}\begin{cases}\left(\frac{t-k}{(k-r)(t-r)}\right)^{1/2}, & r \leq i \leq k-1, \\[2mm] -\left(\frac{k-r}{(t-k)(t-r)}\right)^{1/2}, & k \leq i \leq t-1.\end{cases}$$

Since $\{Z_i\}$ are independent we obtain

$$\mathbb{E}\left[e^{\lambda A_{k,t}^r}\right] = \prod_{i=r}^{t-1}\mathbb{E}\left[e^{\lambda a_i Z_i}\right] \leq \prod_{i=r}^{t-1}\exp\left(\frac{\sigma^2(\lambda a_i)^2}{2}\right) = \exp\left(\frac{\sigma^2\lambda^2}{2}\sum_{i=r}^{t-1}a_i^2\right).$$

A direct computation yields

$$\sum_{i=r}^{t-1}a_i^2 = \frac{1}{\sigma^2}(k-r)\frac{t-k}{(k-r)(t-r)} + \frac{1}{\sigma^2}(t-k)\frac{k-r}{(t-k)(t-r)} = \frac{1}{\sigma^2}\left(\frac{t-k}{t-r} + \frac{k-r}{t-r}\right) = \frac{1}{\sigma^2}.$$

Therefore, $\mathbb{E}\left[e^{\lambda A_{k,t}^r}\right] \leq \exp(\frac{\lambda^2}{2})$, which shows that $A_{k,t}^r$ are 1-sub-Gaussian. Applying the Chernoff bound, for any $\xi > 0$,

$$\mathbb{P}(A_{k,t}^r \geq \xi) \leq \inf_{\lambda > 0}\exp\left(-\lambda\xi + \frac{\lambda^2}{2}\right) = \exp\left(-\frac{\xi^2}{2}\right),$$

and similarly for $\mathbb{P}(A_{k,t}^r \leq -\xi)$. Consequently for any $\xi > 0$,

$$\mathbb{P}\left(|A_{k,t}^r| \geq \xi\right) \leq 2\exp\left(-\frac{\xi^2}{2}\right). \tag{36}$$

Next, for a fixed $r \geq 1$ and any nonnegative sequence $(\gamma_t^r)_{t>r}$ we have, by the union bound [4],

$$
\mathbb{P}\Big(\exists\, k, t \in \mathbb{N},\ r < k < t :\ |A_{k,t}^r| \geq \gamma_t^r\Big)
$$

$$
\leq \sum_{t=r+2}^{\infty} \sum_{k=r+1}^{t-1} \mathbb{P}\Big(|A_{k,t}^r| \geq \gamma_t^r\Big) \ \leq 2 \sum_{t=r+2}^{\infty} (t-r) \exp\Big(-\frac{(\gamma_t^r)^2}{2}\Big). \tag{37}
$$

Let $\tilde{t} := t - r \in \{1, 2, \dots\}$, and write $\gamma_{\tilde{t}}$ for $\gamma_t^r$. To ensure that the right-hand side of Eq. (37) is at most $\alpha_r$, it is sufficient to choose $(\gamma_{\tilde{t}})_{\tilde{t} \geq 1}$ such that

$$
\tilde{t} \cdot \exp\Big(-\frac{\gamma_{\tilde{t}}^2}{2}\Big) \ \leq\ \tilde{t}^{-2} \frac{3}{\pi^2} \alpha_r,
$$

i.e.,

$$
\gamma_{\tilde{t}} = \sqrt{6 \log \tilde{t} + 2 \log(1/\alpha_r) + 2 \log(\pi^2/3)}.
$$

In terms of $t$, this yields

$$
\gamma_t^r = \sqrt{6 \log(t-r) + 2 \log(1/\alpha_r) + 2 \log(\pi^2/3)}. \tag{38}
$$

With this choice,

$$
\mathbb{P}\Big(\exists\, k, t \in \mathbb{N},\ r < k < t :\ |A_{k,t}^r| \geq \gamma_t^r\Big) \leq 2 \sum_{t=r+2}^{\infty} (t-r) \cdot \frac{3}{\pi^2} \alpha_r\, (t-r)^{-3}
$$

$$
= \frac{6}{\pi^2} \alpha_r \sum_{\tilde{t}=2}^{\infty} \tilde{t}^{-2} \leq \frac{6}{\pi^2} \alpha_r \cdot \frac{\pi^2}{6} = \alpha_r. \tag{39}
$$

Finally, relate $A_{k,t}^r$ to $\hat{D}_{k,t}^r$ and $D_{k,t}^r$. Write

$$
\hat{D}_{k,t}^r = \underbrace{\bigg| \frac{1}{\sigma} \Big(\frac{t-k}{(k-r)(t-r)}\Big)^{1/2} \sum_{i=r}^{k-1} X_i - \frac{1}{\sigma} \Big(\frac{k-r}{(t-k)(t-r)}\Big)^{1/2} \sum_{i=k}^{t-1} X_i \bigg|}_{=:B_{k,t}^r},
$$

and

$$
D_{k,t}^r = \underbrace{\bigg| \frac{1}{\sigma} \Big(\frac{t-k}{(k-r)(t-r)}\Big)^{1/2} \sum_{i=r}^{k-1} \mu_i - \frac{1}{\sigma} \Big(\frac{k-r}{(t-k)(t-r)}\Big)^{1/2} \sum_{i=k}^{t-1} \mu_i \bigg|}_{=:b_{k,t}^r}.
$$

Then

$$
B_{k,t}^r - b_{k,t}^r = \frac{1}{\sigma} \Big(\frac{t-k}{(k-r)(t-r)}\Big)^{1/2} \sum_{i=r}^{k-1} (X_i - \mu_i) - \frac{1}{\sigma} \Big(\frac{k-r}{(t-k)(t-r)}\Big)^{1/2} \sum_{i=k}^{t-1} (X_i - \mu_i) = A_{k,t}^r.
$$

Using the inverse triangle inequality,

$$
\big| \hat{D}_{k,t}^r - D_{k,t}^r \big| = \big| |B_{k,t}^r| - |b_{k,t}^r| \big| \leq |B_{k,t}^r - b_{k,t}^r| = |A_{k,t}^r|.
$$

Thus,

$$
\mathbb{P}\Big(\exists\, k, t \in [T],\ r < k < t :\ \big| \hat{D}_{k,t}^r - D_{k,t}^r \big| \geq \gamma_t^r\Big) \ \leq\ \mathbb{P}\Big(\exists\, k, t \in \mathbb{N},\ r < k < t :\ |A_{k,t}^r| \geq \gamma_t^r\Big) \ \leq\ \alpha_r, \tag{40}
$$

where we used Eq. (39) in the last step. $\qquad \square$

---

[4]A more refined peeling argument could sharpen the constants, but it would not change the logarithmic scaling of the threshold.

A.4.3. PROOF OF LEMMA A.3

*Proof.* Recall the bias term

$$\mathcal{R}_T^{\text{bias}} = 2\,\mathbb{E}\left[\sum_{j=1}^{S}\sum_{t=\tau_j}^{\tau_{j+1}-1}\left(\frac{a_L^{(j)}}{a_L^{(j)}+b_t}\right)^2 (\Delta_j^{(\text{eff})})^2 \cdot \mathbf{1}\{N_G^{r(j)} > t\}\right] \tag{41}$$

The proof consists of three steps.

**Step 1 - Decomposition into deterministic and stochastic terms.** First define, for each $t \in \mathcal{I}_j$,

$$E_t(r(j)) := \left\{\forall k \in \{r(j)+1,\ldots,t-1\}:\ \hat{D}_{k,t}^{r(j)} < \gamma_t^{r(j)}\right\}.$$

By definition of the stopping time

$$N_G^{r(j)} = \inf\{t > r(j):\ C_t^{r(j)} \geq \gamma_t^{r(j)}\}, \qquad C_t^{r(j)} := \max_{r(j)<k<t}\hat{D}_{k,t}^{r(j)},$$

we have

$$\{N_G^{r(j)} > t\} \subseteq E_t(r(j)), \qquad t > r(j).$$

That is, if the alarm was not raised by time $t$, then none of the split-point statistics $\hat{D}_{k,t}^{r(j)}$ exceeds the threshold at time $t$. For $t \in \mathcal{I}_j$, define

$$\mathcal{E}_t(r(j)) := \left\{\exists k \in \{r(j)+1,\ldots,t-1\}:\ \hat{D}_{k,t}^{r(j)} < D_{k,t}^{r(j)} - \gamma_t^{r(j)}\right\}, \tag{42}$$

$$H_t(r(j)) := \left\{\forall k \in \{r(j)+1,\ldots,t-1\}:\ D_{k,t}^{r(j)} < 2\gamma_t^{r(j)}\right\}. \tag{43}$$

For every $t \in \mathcal{I}_j$,

$$E_t(r(j)) \subseteq \mathcal{E}_t(r(j)) \cup H_t(r(j)). \tag{44}$$

The idea here is that if an alarm is not raised at time $t$, this can only be due to one of two mechanisms. Either stochastic fluctuations cause the empirical statistic to underestimate the population statistic by at least $\gamma_t^{r(j)}$ (a rare "bad" event), or the population statistic itself is upper bounded by $2\gamma_t^{r(j)}$, which we will later show implies a small contribution to the regret. Mathematically, to establish this inclusion, suppose $E_t(r(j))$ holds but $H_t(r(j))$ fails. Then there is some $\tilde{k} \in (r(j),t)$ with $D_{\tilde{k},t}^{r(j)} \geq 2\gamma_t^{r(j)}$, while $E_t(r(j))$ implies $\hat{D}_{\tilde{k},t}^{r(j)} < \gamma_t^{r(j)}$, so

$$D_{\tilde{k},t}^{r(j)} - \hat{D}_{\tilde{k},t}^{r(j)} > D_{\tilde{k},t}^{r(j)} - \gamma_t^{r(j)} \geq 2\gamma_t^{r(j)} - \gamma_t^{r(j)} = \gamma_t^{r(j)},$$

i.e., $\hat{D}_{\tilde{k},t}^{r(j)} < D_{\tilde{k},t}^{r(j)} - \gamma_t^{r(j)}$, and therefore $\mathcal{E}_t(r(j))$ holds.

Substituting Eq. (44) into the corresponding term in Eq. (41) and using $\left(\frac{a_L^{(j)}}{a_L^{(j)}+b_t}\right)^2 \leq 1$, we obtain

$$(\Delta_j^{(\text{eff})})^2 \sum_{t=\tau_j}^{\tau_{j+1}-1}\left(\frac{a_L^{(j)}}{a_L^{(j)}+b_t}\right)^2 \mathbf{1}\{N_G^{r(j)} > t\}$$

$$\leq (\Delta_j^{(\text{eff})})^2 \sum_{t=\tau_j}^{\tau_{j+1}-1}\left(\frac{a_L^{(j)}}{a_L^{(j)}+b_t}\right)^2 \mathbf{1}\{H_t(r(j))\} + (\Delta_j^{(\text{eff})})^2 \sum_{t=\tau_j}^{\tau_{j+1}-1}\mathbf{1}\{\mathcal{E}_t(r(j))\}. \tag{45}$$

The first term is deterministic for a fixed $r(j)$ and will be bounded by analyzing the population statistic $D_{k,t}^{r(j)}$. To cope with the randomness of $r(j)$ we will upper bound the term inside the expectation path-wise by a quantity that does not depend on

$r(j)$, which will allow us to remove the expectation operator. The second term is stochastic, capturing deviations of $\hat{D}_{k,t}^{r(j)}$ from $D_{k,t}^{r(j)}$, and will be bounded using Lemma A.2. We therefore obtain:

$$\mathcal{R}_T^{\text{bias}} \leq 2\,\mathbb{E}\left[\sum_{j=1}^{S}(\Delta_j^{(\text{eff})})^2 \sum_{t=\tau_j}^{\tau_{j+1}-1} \left(\frac{a_L^{(j)}}{a_L^{(j)}+b_t}\right)^2 \mathbf{1}\{H_t(r(j))\} \right.$$
$$\left. + \sum_{j=1}^{S}(\Delta_j^{(\text{eff})})^2 \sum_{t=\tau_j}^{\tau_{j+1}-1} \mathbf{1}\{\mathcal{E}_t(r(j))\} \right]. \tag{46}$$

**Step 2 - Bounding the deterministic term.** This step highlights the main novelty of our analysis in explicitly accounting for prior missed detections. We control both the bias regret and the detection power in the presence of such missed detections. Their impact is captured by $\Delta_j^{(\text{eff})}$ and $a_L^{(j)}$, which may aggregate weighted contributions from earlier, undetected segments. The resulting bounds remain tractable despite this confounding.

In light of this, we bound the first term inside the expectation in Eq. (46) path-wise for each $j$, i.e., we bound

$$(\Delta_j^{(\text{eff})})^2 \sum_{t=\tau_j}^{\tau_{j+1}-1} \left(\frac{a_L^{(j)}}{a_L^{(j)}+b_t}\right)^2 \mathbf{1}\{H_t(r(j))\}. \tag{47}$$

Define the set of times

$$Y_j := \{t \in \{\tau_j, \tau_j + 1, \ldots, \tau_{j+1} - 1\} : H_t(r(j)) \text{ holds}\}.$$

If $Y_j = \emptyset$ then Eq. (47) is zero; hence assume $Y_j \neq \emptyset$. Let

$$t_{\max} := \max Y_j \quad \text{(largest time in the segment where } H_t(r(j)) \text{ holds),}$$

and let $B_j$ be the corresponding $b_t$ value (i.e., the number of post-change samples)

$$B_j := b_{t_{\max}} = t_{\max} - \tau_j.$$

Then $Y_j \subseteq \{\tau_j, \tau_j + 1, \ldots, t_{\max}\}$, so the set of $b_t$ for which $\mathbf{1}\{H_t(r(j))\} = 1$ is a subset of $\{0, \ldots, B_j\}$. Using the integral test [5] we get

$$\sum_{t \in Y_j} \left(\frac{a_L^{(j)}}{a_L^{(j)}+b_t}\right)^2 \leq \sum_{b=0}^{B_j} \left(\frac{a_L^{(j)}}{a_L^{(j)}+b}\right)^2 \leq 1 + \frac{a_L^{(j)} B_j}{a_L^{(j)}+B_j}, \tag{48}$$

therefore Eq. (47) is upper bounded by

$$(\Delta_j^{(\text{eff})})^2 \sum_{t=\tau_j}^{\tau_{j+1}-1} \left(\frac{a_L^{(j)}}{a_L^{(j)}+b_t}\right)^2 \mathbf{1}\{H_t(r(j))\} \leq (\Delta_j^{(\text{eff})})^2 + (\Delta_j^{(\text{eff})})^2 \frac{a_L^{(j)} B_j}{a_L^{(j)}+B_j}. \tag{49}$$

If $B_j = 0$, then $t_{\max} = \tau_j$ and $Y_j \subseteq \{\tau_j\}$. In this case Eq. (49) gives directly

$$\left(\Delta_j^{(\text{eff})}\right)^2 \sum_{t=\tau_j}^{\tau_{j+1}-1} \left(\frac{a_L^{(j)}}{a_L^{(j)}+b_t}\right)^2 \mathbf{1}\{H_t(r(j))\} \leq \left(\Delta_j^{(\text{eff})}\right)^2 \leq M^2.$$

This is already bounded by the desired bound. Hence, in the remainder of the argument, we may assume $B_j \geq 1$. Then $t_{\max} > \tau_j$, and the split $k^\star = \tau_j$ is admissible at time $t_{\max}$. The final step is to relate the above bias regret term to the ATC detection statistic, which allows us to control the bias through the detection threshold. Let $k^\star = \tau_j$ be the split point [6]

---

[5] Specifically, we use $\sum_{b=1}^{m} \frac{a^2}{(a+b)^2} \leq \int_0^m \frac{a^2}{(a+x)^2}dx = \frac{am}{a+m}$.

[6] The contribution from the edge point $b_t = 0$, corresponding to $t = \tau_j$, is accounted for by the first term in Eq. (49). The CUSUM comparison below is used only when $B_j \geq 1$, so that $k^\star = \tau_j < t_{\max}$.

aligned with the true change point $\tau_j$, comparing samples drawn on opposite sides of the change and thus reflecting the intrinsic signal induced by the shift [7]. The statistic at the split $k^\star = \tau_j$ and time $t_{\max}$ is

$$(D_{k^\star, t_{\max}}^{r(j)})^2 = \frac{1}{\sigma^2} \frac{a_L^{(j)} B_j}{a_L^{(j)} + B_j} (\Delta_j^{(\mathrm{eff})})^2, \tag{50}$$

therefore, Eq. (49) is further upper bounded by:

$$(\Delta_j^{(\mathrm{eff})})^2 \sum_{t=\tau_j}^{\tau_{j+1}-1} \left( \frac{a_L^{(j)}}{a_L^{(j)} + b_t} \right)^2 \mathbf{1}\{H_t(r(j))\} \leq (\Delta_j^{(\mathrm{eff})})^2 + \sigma^2 (D_{k^\star, t_{\max}}^{r(j)})^2. \tag{51}$$

Note that $Y_j \subseteq \{t: D_{k^\star, t}^{r(j)} < 2\gamma_t^{r(j)}\}$, and since $t_{\max} \in Y_j$, we have $D_{k^\star, t_{\max}}^{r(j)} < 2\gamma_{t_{\max}}^{r(j)}$ [8], hence

$$(\Delta_j^{(\mathrm{eff})})^2 \sum_{t=\tau_j}^{\tau_{j+1}-1} \left( \frac{a_L^{(j)}}{a_L^{(j)} + b_t} \right)^2 \mathbf{1}\{H_t(r(j))\} \leq (\Delta_j^{(\mathrm{eff})})^2 + 4\sigma^2 (\gamma_{t_{\max}}^{r(j)})^2 \tag{52}$$

Finally, we bound the sum involving $\gamma_t^{r(j)}$. Recall that

$$(\gamma_t^{r(j)})^2 = \left( 6\log(t - r(j)) + 2\log(1/\alpha_{r(j)}) + 2\log(\pi^2/3) \right).$$

For $t \in \{\tau_j, \ldots, \tau_{j+1} - 1\}$ we have $t - r(j) \leq \tau_{j+1} - r(j)$, so for some absolute constant $C_\gamma > 0$,

$$(\gamma_t^{r(j)})^2 \leq C_\gamma \log\left( \frac{\tau_{j+1} - r(j) + 1}{\alpha_{r(j)}} \right), \qquad t = \tau_j, \ldots, \tau_{j+1} - 1. \tag{53}$$

Combining this with Eq. (52), we conclude that there exists an absolute constant $\tilde{C}_B > 0$ such that

$$(\Delta_j^{(\mathrm{eff})})^2 \sum_{t=\tau_j}^{\tau_{j+1}-1} \left( \frac{a_L^{(j)}}{a_L^{(j)} + b_t} \right)^2 \mathbf{1}\{H_t(r(j))\} \leq (\Delta_j^{(\mathrm{eff})})^2 + \tilde{C}_B \sigma^2 \log\left( \frac{\tau_{j+1} - r(j) + 1}{\alpha_{r(j)}} \right)$$

$$\leq M^2 + \tilde{C}_B \sigma^2 \log\left( \frac{\tau_{j+1}}{\alpha_{r(j)}} \right), \tag{54}$$

where in the last inequality we used $\tau_{j+1} - r(j) \leq \tau_{j+1}$ and $r(j) \geq 1$. This bounds the deterministic CUSUM component of the bias regret for change $j$ from restart $r(j)$. We therefore have:

$$\sum_{j=1}^{S} (\Delta_j^{(\mathrm{eff})})^2 \sum_{t=\tau_j}^{\tau_{j+1}-1} \left( \frac{a_L^{(j)}}{a_L^{(j)} + b_t} \right)^2 \mathbf{1}\{H_t(r(j))\} \leq \sum_{j=1}^{S} \left( M^2 + \tilde{C}_B \sigma^2 \log\left( \frac{\tau_{j+1}}{\alpha_{r(j)}} \right) \right).$$

Since $1 \leq r(j) < \tau_j < \tau_{j+1} \leq T + 1$, and recall that $\alpha_r = \frac{6\alpha}{\pi^2 r^2}$, so $\alpha_{r(j)} \geq c\alpha/T^2$ for all $j$ (for some absolute $c > 0$). Therefore, there exists an absolute constant $C_\alpha > 0$ such that

$$\log\left( \frac{\tau_{j+1}}{\alpha_{r(j)}} \right) \leq C_\alpha \log\left( \frac{T}{\alpha} \right), \qquad j = 1, \ldots, S.$$

Thus, pathwise, and denoting $C_B := \tilde{C}_B \cdot C_\alpha$

$$\sum_{j=1}^{S} \left( M^2 + \tilde{C}_B \sigma^2 \log\left( \frac{\tau_{j+1}}{\alpha_{r(j)}} \right) \right) \leq M^2 S + C_B \sigma^2 S \log\left( \frac{T}{\alpha} \right). \tag{55}$$

Taking expectations preserves this upper bound and concludes the contribution of the deterministic bias term to the overall dynamic regret.

---

[7]If $\tau_{j-1}$ was detected, the split $k^\star = \tau_j$ maximizes the population contrast between the two segments. When earlier changes are missed and the effective pre-change mean becomes a mixture, other split points may yield larger contrasts; nevertheless, the split at $k^\star = \tau_j$ remains a natural and interpretable reference for assessing the detectability of the change and bounding its contribution to the regret.

[8]These steps rely on arguments similar to those in Proposition 3.1. For clarity, we present the derivation explicitly here rather than invoking Proposition 3.1 directly.

**Step 3 - Bounding the stochastic term.** We now bound the second term in Eq. (46). Using $\Delta_j^{(\text{eff})} \leq \Delta_{\text{diam}} \leq M$,

$$\mathbb{E}\left[\sum_{j=1}^{S}\left(\Delta_j^{(\text{eff})}\right)^2 \sum_{t=\tau_j}^{\tau_{j+1}-1} \mathbf{1}\{\mathcal{E}_t(r(j))\}\right] \leq M^2 \sum_{j=1}^{S} \sum_{t=\tau_j}^{\tau_{j+1}-1} \mathbb{P}(\mathcal{E}_t(r(j))). \tag{56}$$

Define

$$\mathcal{E}_t^{\text{abs}}(r) := \left\{\exists k \in \{r+1, \ldots, t-1\} : |\hat{D}_{k,t}^r - D_{k,t}^r| \geq \gamma_t^r\right\}.$$

Since $\mathcal{E}_t(r) \subseteq \mathcal{E}_t^{\text{abs}}(r)$, we have

$$\sum_{j=1}^{S} \sum_{t=\tau_j}^{\tau_{j+1}-1} \mathbb{P}(\mathcal{E}_t(r(j))) \leq \sum_{j=1}^{S} \sum_{t=\tau_j}^{\tau_{j+1}-1} \mathbb{P}(\mathcal{E}_t^{\text{abs}}(r(j))).$$

We do not apply Lemma A.2 conditionally on the random restart index $r(j)$. Instead, we use the following pathwise domination by events indexed by deterministic restart times. For each fixed $t$, the (random) index $r(j)$ appearing above satisfies $1 \leq r(j) \leq t-1$, hence pathwise

$$\mathbf{1}\{\mathcal{E}_t^{\text{abs}}(r(j))\} \leq \sum_{r=1}^{t-1} \mathbf{1}\{\mathcal{E}_t^{\text{abs}}(r)\},$$

and therefore

$$\mathbb{P}(\mathcal{E}_t^{\text{abs}}(r(j))) \leq \sum_{r=1}^{t-1} \mathbb{P}(\mathcal{E}_t^{\text{abs}}(r)).$$

Summing over $t$ and using that $\bigcup_{j=1}^{S}[\tau_j, \tau_{j+1}) \subseteq \{2, \ldots, T\}$, we obtain

$$\sum_{j=1}^{S} \sum_{t=\tau_j}^{\tau_{j+1}-1} \mathbb{P}(\mathcal{E}_t^{\text{abs}}(r(j))) \leq \sum_{t=2}^{T} \sum_{r=1}^{t-1} \mathbb{P}(\mathcal{E}_t^{\text{abs}}(r)).$$

Finally, for fixed $(r, t)$, the same calculation as in the proof of Lemma A.2 yields

$$\mathbb{P}(\mathcal{E}_t^{\text{abs}}(r)) \leq 2(t-r)\exp\left(-\frac{(\gamma_t^r)^2}{2}\right) = \frac{6}{\pi^2}\alpha_r(t-r)^{-2}.$$

Therefore,

$$\sum_{t=2}^{T} \sum_{r=1}^{t-1} \mathbb{P}(\mathcal{E}_t^{\text{abs}}(r)) \leq \sum_{r=1}^{T-1} \sum_{t=r+2}^{\infty} \frac{6}{\pi^2}\alpha_r(t-r)^{-2} \leq \sum_{r=1}^{\infty} \alpha_r = \alpha.$$

Combining with Eq. (56), we obtain

$$\mathbb{E}\left[\sum_{j=1}^{S}\left(\Delta_j^{(\text{eff})}\right)^2 \sum_{t=\tau_j}^{\tau_{j+1}-1} \mathbf{1}\{\mathcal{E}_t(r(j))\}\right] \leq M^2\alpha,$$

and thus its contribution to $\mathcal{R}_T^{\text{bias}}$ is at most $2M^2\alpha$.

Together with Eq. (55), this completes the proof.

$\square$

### A.4.4. PROOF OF LEMMA A.4

*Proof.* In the proof below, we use a pathwise argument to replace the random restart times by a sum over all deterministic restart indices $r \in \{1, \ldots, T\}$. For fixed $r \in \{1, \ldots, T\}$, let $\tau^+(r)$ be the first change after $r$:

$$\tau^+(r) := \min\{t > r : t = \tau_j \text{ for some } j\} \wedge (T+1).$$

Define the false-alarm event

$$\mathsf{FA}_r := \{N_G^r \le \tau^+(r)\}.$$

On the interval $(r, \tau^+(r))$ the mean is constant and hence the population GLR statistic is identically zero: $D_{k,t}^r \equiv 0$ for all $r < k < t \le \tau^+(r)$. If $\mathsf{FA}_r$ occurs, then by definition of $N_G^r$ there exists some $k$ with $r < k < t \le \tau^+(r)$ such that $\hat{D}_{k,t}^r \ge \gamma_t^r$. Therefore

$$\mathsf{FA}_r \subseteq \left\{\exists r < k < t \le \tau^+(r) : \hat{D}_{k,t}^r \ge \gamma_t^r\right\} \subseteq \left\{\exists r < k < t \le T : |\hat{D}_{k,t}^r - D_{k,t}^r| \ge \gamma_t^r\right\}.$$

By Lemma A.2 we thus have, for each $r$,

$$\mathbb{P}(\mathsf{FA}_r) \le \alpha_r.$$

Let $N_{\text{FA}}$ be the number of false alarms produced by the algorithm. Since the restart times $r_m$ are strictly increasing, the algorithm can use each time $r \in \{1, \ldots, T\}$ as a restart at most once, and any false alarm at a restart $r$ implies the event $\mathsf{FA}_r$ occurs. Hence, for every realization,

$$N_{\text{FA}} \le \sum_{r=1}^{T} \mathbf{1}\{\mathsf{FA}_r\}.$$

Using again the fact that $\alpha_r = \frac{6\alpha}{\pi^2 r^2}$ is summable over $r$, we have

$$\mathbb{E}[N_{\text{FA}}] \le \sum_{r=1}^{T} \mathbb{P}(\mathsf{FA}_r) \le \sum_{r=1}^{T} \alpha_r \le \sum_{r=1}^{\infty} \alpha_r \le \alpha.$$

Next, decompose the restarts into detections and false alarms. Each change point can generate at most one detection before the next change (subsequent alarms before the next change point are false alarms), so pathwise

$$K - 1 = \#\{\text{detections}\} + N_{\text{FA}} \le S + N_{\text{FA}},$$

and thus

$$K \le S + 1 + N_{\text{FA}}.$$

Taking expectations and using the bound on $\mathbb{E}[N_{\text{FA}}]$ gives

$$\mathbb{E}[K] \le S + 1 + \mathbb{E}[N_{\text{FA}}] \le S + 1 + \alpha.$$

$\square$

### A.4.5. PROOF OF LEMMA A.5

*Proof.* Let $Z_t := X_t - \mu_t$. By assumption, $(Z_t)_{t \ge 1}$ are independent, mean-zero, and $\sigma^2$-sub-Gaussian. In particular,

$$\mathbb{E}[Z_t^2] \le \sigma^2 \quad \text{and} \quad \mathbb{P}(|Z_t| \ge x) \le 2 \exp\left(-\frac{x^2}{2\sigma^2}\right) \quad \text{for all } x \ge 0. \tag{57}$$

Recall the variance term

$$\mathcal{R}_T^{\text{var}} = 2\mathbb{E}\left[\sum_{m=0}^{K-1} \sum_{t \in B_m} \left(\hat{\mu}_t^{r_m} - \bar{\mu}_t^{r_m}\right)^2\right].$$

For a restart time $r$ and $t > r$, we have

$$\hat{\mu}_t^r - \bar{\mu}_t^r = \frac{1}{t-r} \sum_{i=r}^{t-1} (X_i - \mu_i) = \frac{1}{t-r} \sum_{i=r}^{t-1} Z_i.$$

**Step 1 - A blockwise bound.** Fix a block index $m$ and write $r \equiv r_m$. Let $L_m := |B_m| = r_{m+1} - r_m$ be the block length. For $s \geq 1$, define the partial sum

$$M_s^r := \sum_{i=r}^{r+s-1} Z_i, \qquad \text{so that} \qquad \hat{\mu}_{r+s}^r - \bar{\mu}_{r+s}^r = \frac{1}{s} M_s^r.$$

Condition on $\mathcal{F}_r$ (note that the alarms are stopping times). Then $Z_r$ is $\mathcal{F}_r$-measurable and $(Z_{r+1}, Z_{r+2}, \dots)$ are independent of $\mathcal{F}_r$ with mean zero (Williams, 1991). Hence, for any $s \geq 1$,

$$\mathbb{E}\left[\left(\hat{\mu}_{r+s}^r - \bar{\mu}_{r+s}^r\right)^2 \mid \mathcal{F}_r\right] = \frac{1}{s^2} \mathbb{E}\left[(M_s^r)^2 \mid \mathcal{F}_r\right]$$

$$= \frac{1}{s^2} \left(Z_r^2 + \mathbb{E}\left[\left(\sum_{u=1}^{s-1} Z_{r+u}\right)^2\right]\right)$$

$$= \frac{1}{s^2} \left(Z_r^2 + \sum_{u=1}^{s-1} \mathbb{E}[Z_{r+u}^2]\right) \leq \frac{Z_r^2}{s^2} + \frac{\sigma^2(s-1)}{s^2} \leq \frac{Z_r^2}{s^2} + \frac{\sigma^2}{s}.$$

Using nonnegativity and $L_m \leq T$, we bound

$$\mathbb{E}\left[\sum_{t \in B_m} \left(\hat{\mu}_t^r - \bar{\mu}_t^r\right)^2 \mid \mathcal{F}_r\right] = \mathbb{E}\left[\sum_{s=1}^{L_m} \left(\hat{\mu}_{r+s}^r - \bar{\mu}_{r+s}^r\right)^2 \mid \mathcal{F}_r\right]$$

$$= \sum_{s=1}^{T} \mathbb{E}\left[\left(\hat{\mu}_{r+s}^r - \bar{\mu}_{r+s}^r\right)^2 \mathbf{1}\{L_m \geq s\} \mid \mathcal{F}_r\right]$$

$$\leq \sum_{s=1}^{T} \mathbb{E}\left[\left(\hat{\mu}_{r+s}^r - \bar{\mu}_{r+s}^r\right)^2 \mid \mathcal{F}_r\right]$$

$$\leq Z_r^2 \sum_{s=1}^{\infty} \frac{1}{s^2} + \sigma^2 \sum_{s=1}^{T} \frac{1}{s} \leq \frac{\pi^2}{6} Z_r^2 + \sigma^2(1 + \log T),$$

where we used $\sum_{s \geq 1} s^{-2} = \pi^2/6$ and $H_T = \sum_{s=1}^{T} \frac{1}{s} \leq 1 + \log T$.

Applying the tower property and summing over blocks yields

$$\frac{1}{2} \mathcal{R}_T^{\text{var}} \leq \sigma^2(1 + \log T)\, \mathbb{E}[K] + \frac{\pi^2}{6} \mathbb{E}\left[\sum_{m=0}^{K-1} Z_{r_m}^2\right]. \tag{58}$$

**Step 2 - Controlling the $Z_{r_m}$ term.** For any deterministic $u > 0$, since $\{r_0, \dots, r_{K-1}\} \subseteq \{1, \dots, T\}$, we have pathwise

$$\sum_{m=0}^{K-1} Z_{r_m}^2 \leq \sum_{m=0}^{K-1} \left(u + (Z_{r_m}^2 - u)_+\right) \leq uK + \sum_{t=1}^{T} (Z_t^2 - u)_+.$$

Taking expectations and using Eq. (57),

$$\mathbb{E}\left[(Z_t^2 - u)_+\right] = \int_u^\infty \mathbb{P}(Z_t^2 \geq x)\, dx \leq \int_u^\infty 2\exp\left(-\frac{x}{2\sigma^2}\right) dx = 4\sigma^2 \exp\left(-\frac{u}{2\sigma^2}\right).$$

Therefore,

$$\mathbb{E}\left[\sum_{m=0}^{K-1} Z_{r_m}^2\right] \leq u\, \mathbb{E}[K] + 4\sigma^2 T \exp\left(-\frac{u}{2\sigma^2}\right). \tag{59}$$

Choose $u := 2\sigma^2 \log(eT)$, so that $\exp(-u/(2\sigma^2)) = (eT)^{-1}$ and hence $4\sigma^2 T \exp(-u/(2\sigma^2)) = 4\sigma^2/e$. Plugging into (59) gives

$$\mathbb{E}\left[\sum_{m=0}^{K-1} Z_{r_m}^2\right] \leq 2\sigma^2 \log(eT)\, \mathbb{E}[K] + \frac{4}{e}\sigma^2. \tag{60}$$

**Step 3 - Combining Lemma A.4.**   Combining (58) and (60) yields

$$\mathcal{R}_T^{\mathrm{var}} \leq 2\sigma^2(1 + \log T)\, \mathbb{E}[K] + \frac{\pi^2}{3}\left(2\sigma^2 \log(eT)\, \mathbb{E}[K] + \frac{4}{e}\sigma^2\right).$$

Since $\log(eT) = 1 + \log T$, there exists an absolute constant $C_0 > 0$ such that

$$\mathcal{R}_T^{\mathrm{var}} \leq C_0\, \sigma^2\, (1 + \log T)\, \mathbb{E}[K] + C_0\, \sigma^2.$$

Finally, by Lemma A.4, $\mathbb{E}[K] \leq S + 1 + \alpha$. Since $(S + 1 + \alpha)(1 + \log T) \geq 1$ for $T \geq 2$, we can absorb the additive constant into the main term, yielding (23) for an absolute constant $C_V > 0$.   $\square$

## A.5. Proof of Theorem 4.2

The two main sources of the regret are the variance term (the cumulative estimation error due to the noisy nature of the observations) and the bias term (due to the bias in the estimator after a change, which persists until the policy raises an alarm). In deriving the lower bound, we bound each term separately and then combine the results. To keep notation simple, we prove the lower bound for deterministic policies. The same results also hold for randomized policies by representing the internal random seed of the learner by an auxiliary random variable (independent of the data under every environment), and defining the appropriate filtration. Throughout, we work with an unstructured piecewise-stationary model in which there is no information sharing across segments. Throughout the lower-bound proof we assume $M \geq c_0\sigma$ for a sufficiently large universal constant $c_0$. Under this non-degeneracy condition, all constants below can be chosen universal.

We start with the bias error. For ease of presentation, we first provide the lower bound for the case of a single change point. We then extend in App. A.5.2 to the case of a general number of change points $S$, and in App. A.5.3 we provide the lower bound for the variance error.

### A.5.1. SINGLE CHANGE POINT: BIAS REGRET LOWER BOUND

For lower bounds it suffices to consider a subclass of environments. Fix, without loss of generality $\mu_1 > \mu_0$ and denote $\Delta := |\mu_1 - \mu_0|$ and $A := (\mu_0 + \mu_1)/2$. Consider the Gaussian subclass in which $X_t \sim \mathcal{N}(\mu_t, \sigma^2)$ with piecewise-constant mean $\mu_t$. This subclass satisfies the sub-Gaussian assumption with variance proxy $\sigma^2$. Moreover, when proving a lower bound we may give the learner extra information: in the arguments below we assume the learner knows $\{\mu_0, \mu_1\}$, which can only reduce risk. Hence any lower bound proved in this setting applies to the original problem with unknown means.

Let $\nu_\infty$ denote the no-change instance ($\mu_t = \mu_0\ \forall t \in [T]$), and for $\tau \in [T]$ let $\nu_\tau$ denote the single-change instance with $\mu_t = \mu_0$ for $t < \tau$ and $\mu_t = \mu_1$ for $t \geq \tau$. Write $\mathbb{P}_\infty, \mathbb{E}_\infty$ and $\mathbb{P}_\tau, \mathbb{E}_\tau$ for the corresponding probability laws and expectations. We next lower bound $\mathcal{R}_T(\pi, \nu_\tau)$ on this known-means subclass; since the means are revealed, this lower bound purely captures the bias incurred due to delayed adaptation.

**Lemma A.6.** *For the single-change point case, there exists a universal constant $c > 0$ such that for all $T \geq 3$,*

$$\inf_{\pi \in \Pi}\ \sup_{\tau \in [T] \cup \{\infty\}}\ \mathcal{R}_T(\pi, \nu_\tau)\ \geq\ c\,\sigma^2 \log T.$$

*Proof.* The argument formalizes the stability-adaptivity trade-off. Intuitively, a policy that is too *unstable* incurs regret under $\nu_\infty$ (by frequently behaving as if a change occurred), whereas a policy that is too *stable* cannot reliably react "fast enough" when the change truly happens, and must then incur regret under a suitably chosen $\nu_\tau$.

**Step 1 - Partition of the horizon.**   Fix a window length $1 \leq \ell \leq T$ and define $m := \lfloor T/\ell \rfloor$ disjoint windows

$$n_i := (i-1)\ell + 1, \qquad W_i := \{n_i, n_i + 1, \ldots, n_i + \ell - 1\}, \qquad i = 1, \ldots, m,$$

as illustrated in Figure 11. For each window $W_i$, define the event that the policy "behaves as if the mean is $\mu_1$" for at least half of the window length:

$$A_i := \left\{\hat{\mu}_t > A \text{ for at least } \ell/2 \text{ time steps in } W_i\right\}. \tag{61}$$

This event allows us to lower bound regret in two complementary regimes:

$$\text{On } A_i: \quad \sum_{t \in W_i} (\hat{\mu}_t - \mu_0)^2 \geq \frac{1}{8} \ell \Delta^2, \tag{62}$$

$$\text{On } A_i^c: \quad \sum_{t \in W_i} (\hat{\mu}_t - \mu_1)^2 \geq \frac{1}{8} \ell \Delta^2. \tag{63}$$

Indeed, on $A_i$ there are at least $\ell/2$ indices $t \in W_i$ with $\hat{\mu}_t > A$, hence $\hat{\mu}_t - \mu_0 \geq \Delta/2$ and $(\hat{\mu}_t - \mu_0)^2 \geq \Delta^2/4$ on those indices, which yields $\sum_{t \in W_i} (\hat{\mu}_t - \mu_0)^2 \geq (\ell/2)(\Delta^2/4) = \ell\Delta^2/8$. The bound in Eq. (63) is symmetric.

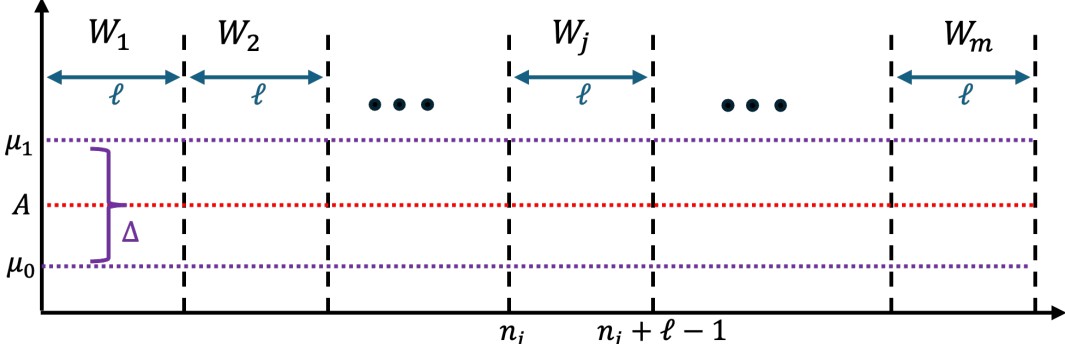

*Figure 11.* Partition of the horizon

**Step 2 - Stability under no change.** We next show that if the policy triggers $A_i$ with non-negligible probability *in every window* under $\nu_\infty$, then it already suffers large regret even when no change occurs. Fix $\alpha \in (0, 1)$ [9]. If $\mathbb{P}_\infty(A_i) > \alpha/m$ for every $i \in [m]$, then as a direct consequence of Eq. (62) we have:

$$\mathcal{R}_T(\pi, \nu_\infty) \geq \sum_{i=1}^m \mathbb{E}_\infty \Big[ \sum_{t \in W_i} (\hat{\mu}_t - \mu_0)^2 \Big] \geq \sum_{i=1}^m \mathbb{P}_\infty(A_i) \cdot \frac{\ell}{8} \Delta^2 > \frac{\alpha}{8} \ell \Delta^2. \tag{64}$$

By Eq. (64), if $\mathbb{P}_\infty(A_i) > \alpha/m$ for all windows, then choosing $\ell = \Theta((\sigma^2 \log T)/\Delta^2)$ yields a regret lower bound of order $\sigma^2 \log T$. Hence, in what follows we restrict attention to policies for which there exists a window index $j \in [m]$ such that

$$q := \mathbb{P}_\infty(A_j) \leq \frac{\alpha}{m}. \tag{65}$$

We view $q$ as quantifying the stability of the policy on window $W_j$ under the "no-change" instance.

**Step 3 - Information constraint (stability limits adaptivity).** We now consider the environment in which the change is at the beginning of this conservative window, i.e., we consider $\nu_{n_j}$. Let

$$p := \mathbb{P}_{n_j}(A_j).$$

On $\nu_{n_j}$, the true mean on $W_j$ is $\mu_1$, hence by Eq. (63),

$$\mathcal{R}_T(\pi, \nu_{n_j}) \geq \mathbb{E}_{n_j} \Big[ \sum_{t \in W_j} (\hat{\mu}_t - \mu_1)^2 \Big] \geq \mathbb{P}_{n_j}(A_j^c) \cdot \frac{\ell}{8} \Delta^2 = (1-p) \cdot \frac{\ell}{8} \Delta^2. \tag{66}$$

Thus, to avoid large regret under $\nu_{n_j}$, the policy must make $p$ close to 1; i.e., it must reliably behave as if the mean is $\mu_1$ within $W_j$. However, since the policy is stable under $\nu_\infty$ on $W_j$ (meaning $q$ is small), distinguishing $\nu_{n_j}$ from $\nu_\infty$ within a window of length $\ell$ requires sufficient statistical information. We now formalize this via the data-processing inequality.

---

[9]Here $\alpha$ is a proof parameter we choose for the lower bound analysis, and not a property of the policy.

**Claim A.7.** *If $\ell$ is chosen so that*

$$\ell \leq \frac{\sigma^2}{\Delta^2} \log \frac{m}{4\alpha}, \tag{67}$$

*then $p \leq 1/2$.*

*Proof.* Since $A_j \in \mathcal{F}_{n_j+\ell-1}$, the data-processing inequality implies

$$D\big(\mathrm{Bern}(p) \,\|\, \mathrm{Bern}(q)\big) \leq D\big(\mathbb{P}_{n_j}^{\leq n_j+\ell-1} \,\|\, \mathbb{P}_{\infty}^{\leq n_j+\ell-1}\big), \tag{68}$$

where $\mathbb{P}^{\leq n}$ denotes the law of the prefix $(X_1, \ldots, X_n)$, and $\mathrm{Bern}(\cdot)$ is the Bernoulli distribution. Under $\nu_{n_j}$ and $\nu_\infty$, the first $n_j - 1$ samples have the same law; the only difference on the prefix up to $n_j + \ell - 1$ is the $\ell$ samples in $W_j$, whose mean is $\mu_1$ under $\nu_{n_j}$ and $\mu_0$ under $\nu_\infty$. Using additivity of KL divergence for independent Gaussians yields

$$D\big(\mathbb{P}_{n_j}^{\leq n_j+\ell-1} \,\|\, \mathbb{P}_{\infty}^{\leq n_j+\ell-1}\big) = \frac{\Delta^2}{2\sigma^2} \, \ell. \tag{69}$$

Combining Eq. (68) and Eq. (69) gives

$$D\big(\mathrm{Bern}(p) \,\|\, \mathrm{Bern}(q)\big) \leq \frac{\Delta^2}{2\sigma^2} \, \ell. \tag{70}$$

We now relate this inequality to the desired conclusion $p \leq 1/2$: For fixed $q \in (0, 1/2]$, the map $p \mapsto D(\mathrm{Bern}(p) \,\|\, \mathrm{Bern}(q))$ is convex, minimized at $p = q$, and non-decreasing on $[1/2, 1]$. Hence, if $p \geq 1/2$, then

$$D(\mathrm{Bern}(p) \,\|\, \mathrm{Bern}(q)) \geq D(\mathrm{Bern}(1/2) \,\|\, \mathrm{Bern}(q)).$$

Using Eq. (65), we have $q \leq \alpha/m \leq 1/2$ (for $m$ large enough), and

$$D\big(\mathrm{Bern}(1/2) \,\|\, \mathrm{Bern}(q)\big) = \frac{1}{2} \log \frac{1}{4q(1-q)} \geq \frac{1}{2} \log \frac{1}{4q} \geq \frac{1}{2} \log \frac{m}{4\alpha}. \tag{71}$$

Therefore, if Eq. (67) holds, then Eq. (70) forces

$$D\big(\mathrm{Bern}(p) \,\|\, \mathrm{Bern}(q)\big) \leq D(\mathrm{Bern}(1/2) \,\|\, \mathrm{Bern}(q)),$$

which implies $p \leq 1/2$. □

Plugging Claim A.7 into Eq. (66) yields

$$\mathcal{R}_T(\pi, \nu_{n_j}) \geq \frac{1}{16} \, \ell \Delta^2. \tag{72}$$

**Step 4 - Choosing $\ell$.** For convenience, fix $\alpha = 1/2$, then Eq. (67) is

$$\ell \leq \frac{\sigma^2}{\Delta^2} \log \frac{m}{2}. \tag{73}$$

Since we are proving a minimax lower bound, it suffices to consider any hard subclass of environments. In particular, we may fix a constant jump size, e.g. $\Delta = 2\sigma$ (equivalently, $\mu_0 = 0$ and $\mu_1 = 2\sigma$), which yields universal constants and ensures the window length below satisfies $1 \leq \ell \leq T$ for all $T \geq 3$.[10] Choose

$$\ell := \left\lceil \frac{\sigma^2}{4\Delta^2} \log T \right\rceil = \left\lceil \frac{1}{16} \log T \right\rceil, \tag{74}$$

and recall $m = \lfloor \frac{T}{\ell} \rfloor \geq \frac{T}{2\ell}$. We have

$$\log \frac{m}{2} \geq \log T - \log(4\ell).$$

---

[10]See Remark A.8 for more details.

With $\ell = \Theta(\log T)$, we have $\log(4\ell) = O(\log \log T)$, and hence

$$\log \frac{m}{2} \geq \log T - O(\log \log T).$$

Therefore, for $T$ large enough (and by adjusting constants for smaller $T$), the choice in Eq. (74) satisfies Eq. (73).

Finally note the following: either Eq. (64) applies, giving $\mathcal{R}_T(\pi, \nu_\infty) \gtrsim \ell \Delta^2 \asymp \sigma^2 \log T$, or there exists $j$ with $q \leq \alpha/m$, in which case Eq. (72) gives $\mathcal{R}_T(\pi, \nu_{n_j}) \gtrsim \ell \Delta^2 \asymp \sigma^2 \log T$. Taking the supremum over $\tau$ and then the infimum over $\pi$ proves Lemma A.6 (absorbing the $O(\log \log T)$ term into constants). $\qquad\square$

*Remark* A.8. In Step 4 we fixed a constant jump size (e.g., $\Delta = 2\sigma$) to keep $\ell = \Theta(\log T)$ and thus ensure $1 \leq \ell \leq T$ and $m = \lfloor T/\ell \rfloor \geq 1$. This is without loss of generality for the minimax lower bound, since restricting to any subclass of environments can only decrease the supremum over $\nu$, and hence any lower bound proved for a fixed $\Delta = \Theta(\sigma)$ applies to the full class. Moreover, unlike other lower bounds (e.g., (Garivier & Moulines, 2008)) where one optimizes a gap parameter to maximize the derived bound, here optimizing over $\Delta$ cannot improve the order of the worst-case bound in $T$, since the information constraint in Claim A.7 depends on the window length only through the KL term $\frac{\Delta^2}{2\sigma^2}\ell$, while the window contribution to squared loss scales as $\ell \Delta^2$. Thus taking $\ell \asymp (\sigma^2/\Delta^2) \log T$ makes the lower bound $\ell \Delta^2$ of order $\sigma^2 \log T$, i.e., the dependence on $\Delta$ cancels up to constants (and lower-order $\log \log T$ terms). Extremely small or extremely large $\Delta$ only make the problem easier (because the two means are nearly indistinguishable in squared loss, or because changes become quickly detectable), so the hardest regime is $\Delta = \Theta(\sigma)$.

### A.5.2. MULTIPLE CHANGE POINTS (BIAS REGRET LOWER BOUND)

We next extend the above results to the multiple change points case. Let $\mathcal{E}_{\leq S,T}(\sigma^2, M)$ be the class of environments with at most $S$ change points[11] $1 = \tau_0 < \tau_1 < \cdots < \tau_S < \tau_{S+1} = T + 1$ with

$$X_t \sim \mathcal{N}(\mu_j, \sigma^2) \qquad \text{for all } t \in [\tau_j, \tau_{j+1}),$$

independently across $t$. Since this is a subclass of the sub-Gaussian model, any lower bound over $\mathcal{E}_{\leq S,T}(\sigma^2, M)$ also applies to the original environment class.

As in Section A.5.1, we prove the bias regret lower bound by restricting $\mathcal{E}_{\leq S,T}(\sigma^2, M)$ to a hard known-means subclass. Specifically, we fix two means $\mu_0 \neq \mu_1$, reveal them to the learner, and consider environments whose means take values in $\{\mu_0, \mu_1\}$ with at most one change per block (hence at most $S$ changes overall). Consider $\Delta$ and $A$ as defined before. We next lower bound $\mathcal{R}_T(\pi, \nu)$ on this class.

**Lemma A.9.** *For the multiple-change point case, there exists a universal constant $c > 0$ such that for all $T \geq 3$,*

$$\inf_{\pi \in \Pi} \sup_{\nu: S(\nu) \leq S} \mathcal{R}_T(\pi, \nu) \geq c \sigma^2 S \log \frac{T}{S}.$$

*Proof.* **Step 1 - Block decomposition.** Fix $S \geq 1$ and define the block length $H := \lfloor T/S \rfloor$. For $s = 1, \ldots, S$, define disjoint blocks

$$B_s := \{(s-1)H + 1, \ldots, sH\}.$$

For each $s$, let $\nu^{(1:s-1)}$ denote a fixed (before observing data) specification of the environment on blocks $B_1, \ldots, B_{s-1}$. We define $\mathbb{P}_s^{\mathrm{nc}}$ (respectively, $\mathbb{E}_s^{\mathrm{nc}}$) as the probability (expectation) law under the continuation environment that agrees with $\nu^{(1:s-1)}$ up to time $(s-1)H$ and has *no change* on block $B_s$, i.e.,

$$\mu_t = \mu_s^{\mathrm{pre}} \text{ for all } t \in B_s,$$

which makes explicit that the probability $\mathbb{P}_s^{\mathrm{nc}}(A_{s,i})$ is computed under the history distribution induced by the previously fixed blocks.

Within each block $B_s$, we further partition time into windows of length $\ell$ (similarly to App. A.5.1):

$$m := \left\lfloor \frac{H}{\ell} \right\rfloor, \qquad \tau_{s,i} := (s-1)H + (i-1)\ell + 1, \qquad W_{s,i} := \{\tau_{s,i}, \ldots, \tau_{s,i} + \ell - 1\}, \qquad i = 1, \ldots, m.$$

---

[11]The construction uses environments with at most $S$ changes, which is sufficient because the minimax risk over the full class is lower bounded by the risk over any subclass.

Next, define the event

$$A_{s,i} := \left\{ (\hat{\mu}_t - A)(\mu_s^{\mathrm{pre}} - A) < 0 \text{ for at least } \ell/2 \text{ time steps in } W_{s,i} \right\}. \tag{75}$$

Thus, $A_{s,i}$ indicates that during at least half of window $W_{s,i}$ the estimator behaves as if the mean has switched to the opposite side of $A$ relative to the pre-block mean. Exactly as in Eq. (62) and Eq. (63), this event implies a window-wise squared-loss lower bound of order $\ell\Delta^2$ under the corresponding (pre- vs post-change) mean.

**Step 2 (Per-block analysis)** Fix $\alpha \in (0,1)$ and consider block $B_s$. First, consider the no-change continuation in which $\mu_t \equiv \mu_s^{\mathrm{pre}}$ throughout $B_s$. If

$$\mathbb{P}_s^{\mathrm{nc}}(A_{s,i}) > \alpha/m \qquad \text{for all } i \in [m] \tag{76}$$

under this "no-change" instance, then exactly as in the single-change analysis (Eq. (64)), the expected loss incurred on block $B_s$ is at least

$$\frac{\alpha}{8}\, \ell\Delta^2. \tag{77}$$

Otherwise, there exists an index $j_s \in [m]$ such that

$$q_s := \mathbb{P}_s^{\mathrm{nc}}(A_{s,j_s}) \leq \alpha/m \tag{78}$$

under the no-change continuation. In this case, we consider the continuation environment in which, on block $B_s$, a change is inserted at time $\tau_{s,j_s}$, flipping the mean from $\mu_s^{\mathrm{pre}}$ to the other value in $\{\mu_0, \mu_1\}$, and keeping the mean constant thereafter on $B_s$. Denote the resulting continuation environment on block $B_s$ by $\nu^{(s)}$ (and let it agree with $\nu^{(1:s-1)}$ on earlier blocks).

Let

$$p_s := \mathbb{P}_{\nu^{(s)}}(A_{s,j_s}).$$

On window $W_{s,j_s}$, the true mean is now the post-change value, and by the same argument as in Eq. (63), the expected loss contributed by $W_{s,j_s}$ is at least

$$(1 - p_s) \cdot \frac{\ell}{8}\Delta^2.$$

**Step 3 - Information-theoretic bound.** Conditioned on the history up to time $\tau_{s,j_s} - 1$, the distributions under the no-change continuation and under $\nu^{(s)}$ differ only on the $\ell$ observations in $W_{s,j_s}$. Therefore, exactly as in Claim A.7, the data-processing inequality yields

$$D(\mathrm{Bern}(p_s) \,\|\, \mathrm{Bern}(q_s)) \leq \frac{\Delta^2}{2\sigma^2}\, \ell.$$

Choosing $\ell$ to satisfy the analogue of Eq. (73) with $m = \lfloor H/\ell \rfloor$ forces $p_s \leq 1/2$, and hence the expected loss on block $B_s$ is at least

$$\frac{1}{16}\, \ell\Delta^2.$$

**Step 4 - Choosing $\ell$.** As in the single change case, take $\alpha = 1/2$ and $\Delta = 2\sigma$, and choose[12]

$$\ell := \left\lceil \frac{\sigma^2}{4\Delta^2} \log H \right\rceil = \left\lceil \frac{1}{16} \log H \right\rceil, \qquad m := \left\lfloor \frac{H}{\ell} \right\rfloor. \tag{79}$$

Since $m = \lfloor H/\ell \rfloor \geq H/(2\ell)$, we have

$$\log m \geq \log H - \log(2\ell) = \log H - O(\log \log H),$$

and therefore the KL condition in Step 3 holds for $H$ large enough (and otherwise can be absorbed into universal constants). Thus, in each block $B_s$, the construction guarantees an expected loss of at least $c_0\, \ell\Delta^2$ for some universal constant $c_0 > 0$, either due to instability under no change or due to detection delay after an inserted change.

---

[12]We focus on the regime where $H = \lfloor T/S \rfloor$ is larger than a sufficiently large universal constant; when $H = O(1)$ (dense-change regime), the variance lower bound in Lemma A.10 already yields the claimed order in Theorem 4.2.

Finally, define a single global environment $\nu$ by specifying it block-by-block as follows: starting from block $s = 1$ and proceeding to $s = S$, given the already fixed behavior on blocks $B_1, \ldots, B_{s-1}$, if the instability condition holds on block $s$ we set block $s$ to have no change; otherwise we insert a change at $\tau_{s,j_s}$ as above. This produces at most one change in each block and therefore at most $S$ change points overall.

Since the blocks are disjoint and losses are nonnegative, summing over $s = 1, \ldots, S$ yields

$$\inf_{\pi \in \Pi} \sup_{\nu:\, S(\nu) \leq S} \mathcal{R}_T(\pi, \nu) \;\geq\; c_0 \, S \, \ell \Delta^2 \;\asymp\; \sigma^2 \, S \log H \;=\; \Omega\!\left( \sigma^2 S \log \frac{T}{S} \right),$$

up to the same lower-order $O(\log\log(T/S))$ effects as in the single-change proof.

$\square$

### A.5.3. VARIANCE REGRET LOWER BOUND

We lower bound the variance contribution (estimation error). We first consider the relaxation in which the learner is told all change points $\tau_1, \ldots, \tau_S$, but the segment means $\{\mu_j\}_{j=0}^S$ are unknown. Conditioned on the change points, the horizon decomposes into $S + 1$ stationary segments with lengths

$$L_j := \tau_{j+1} - \tau_j, \qquad j = 0, \ldots, S, \qquad \text{and} \qquad \sum_{j=0}^{S} L_j = T.$$

Revealing the change points can only help the learner, hence any lower bound for this relaxed problem also applies to the original minimax risk. The lower bound on this class is given in the next lemma.

**Lemma A.10.** *There exists a universal constant $c > 0$ such that for all $T \geq 3$,*

$$\inf_{\pi \in \Pi} \sup_{\{\mu_j\}_{j=0}^S} \mathcal{R}_T(\pi, \nu) \;\geq\; c \sigma^2 (S+1) \left( 1 + \log \frac{T}{S+1} \right), \tag{80}$$

*where $\{\mu_j\}_{j=0}^S$ satisfy Eq. (1).*

*Proof.* **Step 1 - Sequential estimation lower bound.** We first lower bound the variance contribution incurred within a single stationary segment of length $n \geq 2$, in which the mean is constant and equal to $\mu$. That is, the observations are i.i.d.

$$X_1, \ldots, X_n \sim \mathcal{N}(\mu, \sigma^2).$$

Our goal is to obtain a minimax lower bound on the expected cumulative squared estimation error, i.e., on

$$\inf_{\{\hat{\mu}_t\}_{t=1}^n} \sup_{\mu \in [-M/2, M/2]} \mathbb{E}_\mu \left[ \sum_{t=1}^n (\hat{\mu}_t - \mu)^2 \right],$$

where each $\hat{\mu}_t$ is measurable with respect to $\sigma(X_1, \ldots, X_{t-1})$. Throughout this step, we restrict $\mu$ to lie in $[-M/2, M/2]$; this restriction defines a valid subclass of the original problem and will later be used to ensure that the diameter constraint in Eq. (1) is satisfied.

By the Bayes-minimax inequality, for any prior $\rho$ supported on $[-M/2, M/2]$ we have

$$\inf_{\{\hat{\mu}_t\}_{t=1}^n} \sup_{\mu \in [-M/2, M/2]} \mathbb{E}_\mu \left[ \sum_{t=1}^n (\hat{\mu}_t - \mu)^2 \right] \;\geq\; \inf_{\{\hat{\mu}_t\}_{t=1}^n} \mathbb{E}_{\mu \sim \rho} \, \mathbb{E}_\mu \left[ \sum_{t=1}^n (\hat{\mu}_t - \mu)^2 \right].$$

We therefore choose a convenient prior $\rho = \rho_M$ supported on $[-M/2, M/2]$, and show that there exists a constant $c > 0$ (universal when $M \geq c_0 \sigma$) such that

$$\inf_{\{\hat{\mu}_t\}_{t=1}^n} \mathbb{E}_{\mu \sim \rho_M} \, \mathbb{E}_\mu \left[ \sum_{t=1}^n (\hat{\mu}_t - \mu)^2 \right] \;\geq\; c \sigma^2 \log(n+1). \tag{81}$$

**Proof of Eq.** (81). Consider the Bayes model in which $\mu \sim \rho_M$ has density

$$p_M(\mu) = \frac{2}{M} \cos^2\left(\frac{\pi\mu}{M}\right) \mathbf{1}\{|\mu| \leq M/2\},$$

and conditionally on $\mu$ the observations are i.i.d. $X_1, \ldots, X_n \sim \mathcal{N}(\mu, \sigma^2)$. Since $\rho_M$ is supported on $[-M/2, M/2]$ and satisfies $p_M(\pm M/2) = 0$, van Trees applies. Let $I(\rho_M)$ denote the Fisher information of the prior. A direct computation gives

$$\frac{d}{d\mu} \log p_M(\mu) = -\frac{2\pi}{M} \tan\left(\frac{\pi\mu}{M}\right), \qquad I(\rho_M) = \int \left(\frac{p_M'(\mu)}{p_M(\mu)}\right)^2 p_M(\mu)\, d\mu = \frac{4\pi^2}{M^2}.$$

We first write

$$\inf_{\{\hat{\mu}_t\}_{t=1}^n} \mathbb{E}_{\mu \sim \rho_M} \mathbb{E}_\mu \left[\sum_{t=1}^n (\hat{\mu}_t - \mu)^2\right] = \inf_{\{\hat{\mu}_t\}_{t=1}^n} \sum_{t=1}^n \mathbb{E}_{\mu \sim \rho_M} \mathbb{E}_\mu \left[(\hat{\mu}_t - \mu)^2\right].$$

For the Gaussian model, the Fisher information in $t - 1$ samples equals $(t - 1)/\sigma^2$. Thus, for any estimator $\hat{\mu}_t$ based on $t - 1$ samples, van Trees yields

$$\mathbb{E}_{\mu \sim \rho_M} \mathbb{E}_\mu \left[(\hat{\mu}_t - \mu)^2\right] \geq \frac{1}{(t-1)/\sigma^2 + I(\rho_M)} = \frac{\sigma^2}{t - 1 + \sigma^2 I(\rho_M)} = \frac{\sigma^2}{t - 1 + a}, \qquad a := \frac{4\pi^2 \sigma^2}{M^2}.$$

Summing over $t$ and using the harmonic series lower bound gives

$$\inf_{\{\hat{\mu}_t\}_{t=1}^n} \mathbb{E}_{\mu \sim \rho_M} \mathbb{E}_\mu \left[\sum_{t=1}^n (\hat{\mu}_t - \mu)^2\right] \geq \sigma^2 \sum_{t=1}^n \frac{1}{t - 1 + a} \geq \sigma^2 \log\left(\frac{n - 1 + a}{a}\right) = \Omega(\sigma^2 \log(n + 1)).$$

If $M \geq c_0 \sigma$, then $a \leq a_0$ is an absolute constant. By the Bayes–minimax inequality, this implies that

$$\inf_{\{\hat{\mu}_t\}_{t=1}^n} \sup_{\mu \in [-M/2, M/2]} \mathbb{E}_\mu \left[\sum_{t=1}^n (\hat{\mu}_t - \mu)^2\right] \geq c\sigma^2 \log(n + 1).$$

**Step 2 - Summing over segments.** Place an independent product prior on segment means:

$$\mu_0, \ldots, \mu_S \stackrel{\text{i.i.d.}}{\sim} \rho_M, \qquad \text{so that } \mu_j \in [-M/2, M/2] \text{ a.s.}$$

This construction is consistent with the restriction imposed in Step 1 and defines a valid hard subclass satisfying the diameter constraint, since under this prior we have $\max_{u,v} |\mu_u - \mu_v| \leq M$ almost surely.

Recall $\mathcal{I}_j = [\tau_j, \tau_{j+1})$ and let $L_j = |\mathcal{I}_j| = \tau_{j+1} - \tau_j$. For $t \in \mathcal{I}_j$, write $s := t - \tau_j + 1$. At prediction time $t$, the learner has observed only the first $s - 1$ samples from segment $j$. Since the prior factorizes across segments and observations outside segment $j$ are independent of $\mu_j$, data outside segment $j$ carries no information about $\mu_j$. Thus the prediction at time $t$ is no easier than the $s$-th prediction in the single-segment estimation problem of Step 1.

Therefore, applying Step 1 with $n = s \leq L_j$ and summing over segments yields

$$\inf_{\pi \in \Pi} \mathbb{E}\left[\sum_{t=2}^T (\hat{\mu}_t - \mu_t)^2\right] \geq \sigma^2 \sum_{j=0}^S \log\left(\frac{L_j + 1 + a}{1 + a}\right) \geq c_2 \sigma^2 \sum_{j=0}^S \log(L_j + 1),$$

for a constant $c_2 > 0$ that is universal when $M \geq c_0 \sigma$ (since then $a \leq a_0$ is an absolute constant). By the Bayes–minimax inequality, the same lower bound applies to the minimax risk under this fixed segmentation.

**Step 3 - Choosing a hard partition.** Choose change points so that the $S + 1$ segments have (approximately) equal length, e.g. $L_j \in \{\lfloor T/(S+1) \rfloor, \lceil T/(S+1) \rceil\}$ for all $j$. Then $\log(L_j + 1) \geq \log(\lfloor T/(S+1) \rfloor + 1)$ for all $j$, and hence

$$\sum_{j=0}^S \log(L_j + 1) \geq (S + 1) \log\left(\left\lfloor \frac{T}{S+1} \right\rfloor + 1\right) \gtrsim (S + 1)\left(1 + \log \frac{T}{S+1}\right),$$

up to absolute constants. Plugging into the previous display yields Lemma A.10. $\qquad\square$

Since Lemma A.9 is proved on a subclass (revealed means) and Lemma A.10 is proved under a relaxation (revealed change points), both bounds apply to $R_T^\star(\mathcal{E}_{\leq S,T}(\sigma^2, M))$; taking the maximum yields Theorem 4.2.

*Remark* A.11. The argument above provides a minimax lower bound by combining two relaxations, and hence guarantees at least the larger of the detection and estimation costs. Constructing a single instance in which both mechanisms contribute additively is possible but results in the same lower bound rate; thus, we omit it.

### A.6. Proof of Proposition 3.1

We first show a deterministic algebraic inequality. This is the key calculation showing that the loss of SNR caused by using a contaminated pre-change reference is controlled by the population evidence of the previous missed change.

**Claim A.12.** *Fix $n_0, n_1, n_2 > 0$ and three means $\mu_0, \mu_1, \mu_2$. Let*

$$\mu_{01} := \frac{n_0\mu_0 + n_1\mu_1}{n_0 + n_1}.$$

*Define*

$$S^\star := \frac{n_1 n_2}{n_1 + n_2} \frac{(\mu_2 - \mu_1)^2}{\sigma^2},$$

$$S^{\text{eff}} := \frac{(n_0 + n_1)n_2}{(n_0 + n_1) + n_2} \frac{(\mu_2 - \mu_{01})^2}{\sigma^2},$$

*and*

$$S^{\text{prev}} := \frac{n_0 n_1}{n_0 + n_1} \frac{(\mu_1 - \mu_0)^2}{\sigma^2}.$$

*Then*

$$\left(S^\star - S^{\text{eff}}\right)_+ \leq S^{\text{prev}}.$$

*Proof.* It suffices to prove the claim with $\sigma = 1$, since all three quantities are divided by the same factor $\sigma^2$.

Let

$$\delta := \mu_1 - \mu_0, \qquad \Delta := \mu_2 - \mu_1, \qquad n_L := n_0 + n_1, \qquad w := \frac{n_0}{n_L}.$$

Then

$$\mu_{01} = \mu_1 - w\delta, \qquad \mu_2 - \mu_{01} = \Delta + w\delta.$$

Define

$$A := \frac{n_1 n_2}{n_1 + n_2}, \qquad B := \frac{n_L n_2}{n_L + n_2}, \qquad P := \frac{A}{B} \in (0, 1).$$

Thus

$$S^\star = A\Delta^2, \qquad S^{\text{eff}} = B(\Delta + w\delta)^2,$$

and therefore

$$S^\star - S^{\text{eff}} = B\left(P\Delta^2 - (\Delta + w\delta)^2\right).$$

Let $b := w\delta$. Then

$$P\Delta^2 - (\Delta + b)^2 = (P - 1)\Delta^2 - 2b\Delta - b^2.$$

Since $P < 1$, set $c := 1 - P > 0$. Completing the square gives

$$(P - 1)\Delta^2 - 2b\Delta - b^2 = -c\left(\Delta + \frac{b}{c}\right)^2 + \frac{b^2 P}{c} \leq \frac{b^2 P}{1 - P}.$$

Multiplying by $B$ and using $BP = A$, we get

$$S^\star - S^{\text{eff}} \leq \frac{Ab^2}{1 - P}.$$

A direct calculation gives

$$1 - P = 1 - \frac{A}{B} = \frac{n_0 n_2}{(n_0 + n_1)(n_1 + n_2)}.$$

Therefore,

$$\frac{Aw^2}{1 - P} = \frac{\frac{n_1 n_2}{n_1 + n_2} \cdot \frac{n_0^2}{(n_0 + n_1)^2}}{\frac{n_0 n_2}{(n_0 + n_1)(n_1 + n_2)}} = \frac{n_0 n_1}{n_0 + n_1}.$$

Since $b = w\delta$, this implies

$$S^\star - S^{\mathrm{eff}} \leq \frac{n_0 n_1}{n_0 + n_1} \delta^2 = S^{\mathrm{prev}}$$

when $\sigma = 1$. Restoring the common normalization by $\sigma^2$ gives the claim. Finally, applying $(\cdot)_+$ to the left-hand side completes the proof. $\qquad\square$

We now prove Proposition 3.1. Fix a deterministic restart index $r$. Define the fixed-restart confidence event

$$\mathcal{G}_r := \left\{ \forall k, t \in [T] \text{ with } r < k < t : \left| \widehat{D}_{k,t}^r - D_{k,t}^r \right| < \gamma_t^r \right\}.$$

By Lemma A.2,

$$\mathbb{P}_\nu(\mathcal{G}_r) \geq 1 - \alpha_r.$$

We show that on $\mathcal{G}_r$, the claimed implication holds simultaneously for all relevant $j$ and $t$. Fix a change index $j$ such that $r < \tau_{j-1} < \tau_j$, and fix

$$t \in \{\tau_j + 1, \ldots, \tau_{j+1} - 1\}.$$

Assume that $N_G^r > t$. Since $t > \tau_j$, this implies in particular that $N_G^r > \tau_j$. Hence the detector initialized at $r$ has not raised an alarm by time $\tau_j$, and therefore

$$C_{\tau_j}^r < \gamma_{\tau_j}^r.$$

Since $r < \tau_{j-1} < \tau_j$, the split $k = \tau_{j-1}$ is admissible at time $\tau_j$. Thus,

$$\widehat{D}_{\tau_{j-1}, \tau_j}^r \leq C_{\tau_j}^r < \gamma_{\tau_j}^r.$$

On the event $\mathcal{G}_r$, we also have

$$\left| \widehat{D}_{\tau_{j-1}, \tau_j}^r - D_{\tau_{j-1}, \tau_j}^r \right| < \gamma_{\tau_j}^r.$$

Therefore,

$$D_{\tau_{j-1}, \tau_j}^r \leq \widehat{D}_{\tau_{j-1}, \tau_j}^r + \left| \widehat{D}_{\tau_{j-1}, \tau_j}^r - D_{\tau_{j-1}, \tau_j}^r \right| < 2\gamma_{\tau_j}^r,$$

and hence

$$\left( D_{\tau_{j-1}, \tau_j}^r \right)^2 \leq 4(\gamma_{\tau_j}^r)^2.$$

It remains to relate this population CUSUM statistic to the SNR degradation. Let

$$n_0 := \tau_{j-1} - r, \qquad n_1 := \tau_j - \tau_{j-1}, \qquad n_2 := t - \tau_j.$$

Since $t > \tau_j$, we have $n_2 > 0$. Define

$$\mu_0 := \frac{1}{\tau_{j-1} - r} \sum_{i=r}^{\tau_{j-1}-1} \mu_i, \qquad \mu_1 := \mu_{j-1}, \qquad \mu_2 := \mu_j.$$

The effective pre-change mean for detecting $\tau_j$ from restart $r$ is

$$\mu_{01} = \frac{n_0 \mu_0 + n_1 \mu_1}{n_0 + n_1}.$$

With these definitions,

$$\mathrm{SNR}_j^\star(t) = \frac{n_1 n_2}{n_1 + n_2} \frac{(\mu_2 - \mu_1)^2}{\sigma^2},$$

and

$$\mathrm{SNR}_j^{\mathrm{eff}}(t; r) = \frac{(n_0 + n_1)n_2}{(n_0 + n_1) + n_2} \frac{(\mu_2 - \mu_{01})^2}{\sigma^2}.$$

Applying Claim A.12 gives

$$\left(\mathrm{SNR}_j^\star(t) - \mathrm{SNR}_j^{\mathrm{eff}}(t; r)\right)_+ \leq \frac{n_0 n_1}{n_0 + n_1} \frac{(\mu_1 - \mu_0)^2}{\sigma^2}.$$

The right-hand side is exactly the squared population CUSUM statistic at time $\tau_j$ and split $k = \tau_{j-1}$:

$$\frac{n_0 n_1}{n_0 + n_1} \frac{(\mu_1 - \mu_0)^2}{\sigma^2} = \left(D_{\tau_{j-1}, \tau_j}^r\right)^2.$$

Consequently, on $\mathcal{G}_r$, whenever $N_G^r > t$,

$$\left(\mathrm{SNR}_j^\star(t) - \mathrm{SNR}_j^{\mathrm{eff}}(t; r)\right)_+ \leq \left(D_{\tau_{j-1}, \tau_j}^r\right)^2 \leq 4(\gamma_{\tau_j}^r)^2.$$

Finally, by the definition of the threshold,

$$(\gamma_{\tau_j}^r)^2 = 6\log(\tau_j - r) + 2\log(1/\alpha_r) + 2\log(\pi^2/3) \leq C \log\left(\frac{\tau_j - r + 1}{\alpha_r}\right)$$

for a universal constant $C > 0$. Therefore, on $\mathcal{G}_r$,

$$N_G^r > t \implies \left(\mathrm{SNR}_j^\star(t) - \mathrm{SNR}_j^{\mathrm{eff}}(t; r)\right)_+ \leq C \log\left(\frac{\tau_j - r + 1}{\alpha_r}\right).$$

Since $\mathbb{P}_\nu(\mathcal{G}_r) \geq 1 - \alpha_r$, the proof is complete.

## A.7. Extension to vector-valued observations

This subsection summarizes how the ATC algorithm and the upper-bound analysis extend to the case $X_t \in \mathbb{R}^d$ with piecewise-constant mean $\mu_t \in \mathbb{R}^d$. We focus on the key changes (statistic, threshold, confidence bound, and the resulting bias/variance scaling), and omit the full proofs.

**Vector-valued model and regret.** Assume the same change-point structure as in Section 2, except that $X_t, \mu_t \in \mathbb{R}^d$. We measure performance by the squared Euclidean dynamic regret

$$\mathcal{R}_T(\pi, \nu) = \mathbb{E}_{\pi, \nu}\left[\sum_{t=2}^T \|\hat{\mu}_t^\pi - \mu_t\|_2^2\right]. \tag{82}$$

We assume an isotropic vector sub-Gaussian noise model with proxy $\sigma^2$: for all $t \geq 1$, all $\lambda \in \mathbb{R}$, and all $u \in \mathbb{R}^d$ with $\|u\|_2 = 1$,

$$\mathbb{E}\left[\exp\left(\lambda u^\top (X_t - \mu_t)\right)\right] \leq \exp\left(\frac{\sigma^2 \lambda^2}{2}\right). \tag{83}$$

**Algorithmic structure.** ATC maintains restart times $r$ and outputs the running mean since the last restart:

$$\hat{\mu}_t = \frac{1}{t - r} \sum_{i=r}^{t-1} X_i \in \mathbb{R}^d. \tag{84}$$

Implementation uses vector prefix sums $G_t = \sum_{i=1}^t X_i \in \mathbb{R}^d$, so that $\hat{\mu}_t = (G_{t-1} - G_{r-1})/(t - r)$.

**Vector CUSUM/GLR statistic.** For $r < k < t$, define the block averages

$$\bar{X}_{r:k-1} := \frac{1}{k-r}\sum_{i=r}^{k-1}X_i, \qquad \bar{X}_{k:t-1} := \frac{1}{t-k}\sum_{i=k}^{t-1}X_i,$$

and the (standardized) two-sample mean-difference statistic

$$\hat{D}_{k,t}^r := \frac{1}{\sigma}\sqrt{\frac{(k-r)(t-k)}{(t-r)}}\,\big\|\bar{X}_{r:k-1}-\bar{X}_{k:t-1}\big\|_2. \tag{85}$$

The scan statistic and stopping time remain

$$C_t^r := \max_{r<k<t}\hat{D}_{k,t}^r, \qquad N_G^r := \inf\{t > r :\ C_t^r \geq \gamma_t^r\}.$$

Computationally, each candidate evaluation is now $O(d)$, so the naive scan is $O(d(t-r))$ per time step, and the multiscale approximation yields $O(d\log(t-r))$ candidates per time step.

**Population statistic and effective gaps.** Define the population analogue

$$D_{k,t}^r := \frac{1}{\sigma}\sqrt{\frac{(k-r)(t-k)}{(t-r)}}\,\big\|\bar{\mu}_{r:k-1}-\bar{\mu}_{k:t-1}\big\|_2. \tag{86}$$

All "effective mean" definitions in Appendix A.4.1 extend by replacing absolute values by $\|\cdot\|_2$; in particular, for a change $\tau_j$ inside a block started at $r$, the effective gap becomes

$$\Delta_j^{(\mathrm{eff})} := \big\|\mu_L^{(j)}-\mu_j\big\|_2. \tag{87}$$

**Confidence bound and threshold.** The key modification is a uniform concentration inequality for the vector statistic. Let $A_{k,t}^r \in \mathbb{R}^d$ be the vector analogue of App. A.4.2:

$$A_{k,t}^r := \frac{1}{\sigma}\left(\frac{t-k}{(k-r)(t-r)}\right)^{1/2}\sum_{i=r}^{k-1}(X_i-\mu_i)\ -\ \frac{1}{\sigma}\left(\frac{k-r}{(t-k)(t-r)}\right)^{1/2}\sum_{i=k}^{t-1}(X_i-\mu_i).$$

Then by the reverse triangle inequality,

$$\big|\hat{D}_{k,t}^r - D_{k,t}^r\big| \leq \|A_{k,t}^r\|_2. \tag{88}$$

Under (83), $A_{k,t}^r$ is a 1-sub-Gaussian vector (in the standard sense that all one-dimensional projections are $\sigma^2$-sub-Gaussian), which implies a tail bound of the form $\mathbb{P}(\|A_{k,t}^r\|_2 \geq C(\sqrt{d}+\sqrt{x})) \leq e^{-x}$ (for some universal constant C). Combining this tail bound with the same union bound over $(k,t)$ as in App. A.4.2 yields the following threshold:

$$\gamma_t^r := C_0\left(\sqrt{d}+\sqrt{6\log(t-r)+2\log(1/\alpha_r)+2\log(\pi^2/3)}\right), \qquad \alpha_r = \frac{6}{\pi^2}\frac{\alpha}{r^2}, \tag{89}$$

for some universal constant $C_0 > 0$. Following the same arguments as in App. A.4.2, we can show that for each fixed deterministic $r$, with $\gamma_t^r$ as in Eq. (89),

$$\mathbb{P}\Big(\exists k,t \in [T],\ r < k < t :\ \big|\hat{D}_{k,t}^r - D_{k,t}^r\big| \geq \gamma_t^r\Big) \leq \alpha_r. \tag{90}$$

As in the scalar case, this implies $\mathbb{E}[N_{\mathrm{FA}}] \leq \sum_r \alpha_r \leq \alpha$ and hence $\mathbb{E}[K] \leq S+1+\alpha$ (where $K$ is the number of restart blocks).

**Effect on variance regret.** Within a block, $\hat{\mu}_t - \bar{\mu}_t^r$ is an average of independent centered sub-Gaussian vectors. Under Eq. (83), one obtains a pointwise bound

$$\mathbb{E}\big[\|\hat{\mu}_t - \bar{\mu}_t^r\|_2^2\big] \leq c\,\frac{d\sigma^2}{t-r}, \tag{91}$$

for a universal constant $c > 0$. Summing over time within each block (using arguments similar to those in the $d = 1$ case) yields harmonic series terms as in App. A.4.5, and using $\mathbb{E}[K] \leq S+1+\alpha$ gives

$$\mathcal{R}_T^{\mathrm{var}} \leq C_V^{(d)}\,d\sigma^2\,(S+1+\alpha)\,(1+\log T), \tag{92}$$

for some absolute constant $C_V^{(d)} > 0$ (absorbing universal constants).

**Effect on bias regret.** The regret decomposition in Lemma A.1 carries over with $(\cdot)^2$ replaced by $\|\cdot\|_2^2$ and effective gaps as in Eq. (87). The deviation event contribution is controlled as before, using the vector confidence event $M^2\alpha$ (where here $M$ is defined as the natural extension to the $d$-dimensional setting). The deterministic drift-threshold argument is unchanged except that it now compares $D_{k,t}^r$ to the larger threshold in Eq. (89). Since $(\gamma_t^r)^2 \lesssim d + \log((t-r)/\alpha_r)$, the corresponding regret contribution scales as $\sigma^2(\gamma_t^r)^2 \lesssim \sigma^2(d + \log((t-r)/\alpha_r))$ (beyond the scalar $O(\sigma^2 \log(T/\alpha))$ term). Concretely, for some absolute constant $C_B^{(d)} > 0$,

$$\mathcal{R}_T^{\text{bias}} \;\le\; M^2\,(\alpha + S) \;+\; C_B^{(d)}\,\sigma^2\,S\left(d + \log\frac{T}{\alpha}\right). \tag{93}$$

**Resulting regret upper bound in $\mathbb{R}^d$.** Combining Eq. (92) and Eq. (93) yields the vector analogue of Theorem 4.1: there exist constants $C_V^{(d)}, C_B^{(d)} > 0$ such that

$$\mathcal{R}_T^{\text{ATC}}(d) \;\le\; C_V^{(d)}\,d\sigma^2\,(S + 1 + \alpha)\,(1 + \log T) \;+\; C_B^{(d)}\,\sigma^2\,S\left(d + \log\frac{T}{\alpha}\right) \;+\; M^2\,(\alpha + S). \tag{94}$$

For fixed $\alpha \in (0, 1)$, the dominant scaling is $\mathcal{R}_T^{\text{ATC}} = O\big(d\sigma^2(S+1)\log(T)\big)$, up to constants.

