# OpenReview forum: "The Cost of Learning Under Multiple Change Points"
_ICML.cc/2026/Conference — ICML 2026 regular_

### Official Review · Reviewer_5UDN · 2026-03-06

**Soundness:** 2
**Presentation:** 2
**Significance:** 2
**Originality:** 3
**Overall Recommendation:** 4
**Confidence:** 2

**Summary:**

In this work, the authors investigate an online learning problem, in which the goal of the learner is to accurately estimate the mean of a stream of sequentially observed independent $\sigma^{2}$-sub-Gaussian samples. The mean of these $\sigma^{2}$-sub-Gaussian samples is piecewise stationary, which means that the mean changes at multiple time-steps (referred to as change points) over a finite horizon, and remains stationary in between. The number of changes and the horizon are assumed to be unknown, while the learner knows the $\sigma^{2}$-sub-Gaussianity of the samples. The multiple change points in the environment may causes endogenous confounding, which refers to the scenarios under which one missed change in the past makes all estimates inaccurate in the future. To avoid this issue, the methods in prior works are said to require assumptions on either the minimum magnitude in mean shifts or the minimum spacing between change points. Unlike these methods, the authors proposed the Anytime Tracking CUSUM (ATC) algorithm that achieves nearly minimax-optimal cumulative squared error (between the estimate and the true mean) without imposing similar assumptions. Some experiments on both synthetic and real-world dataset demonstrate the good performance of this algorithm in practice.

**Compliance With Llm Reviewing Policy:**

Affirmed.

**Final Justification:**

After discussing with the authors, I will raise my score due to their efforts in their response.

**Key Questions For Authors:**

1. Following the first weakness, what is the connection between the cumulative squared error in this work and the regret in the non-stationary online learning problems? Why do you choose this metric instead of using those in the prior works?
2. Following the first question, how would you design the sampling algorithm when considering the non-stationary online learning problem in prior works?
3. Following the second weakness, what is the probability of the upper bound in Lemma 3.1 being true?

**Limitations:**

Yes, the authors mentioned that their performance cannot be extended to other notions of regret. In addition, this work cannot be extended to non-stationary online decision making problem, which are the ones with detectability assumption.

**Strengths And Weaknesses:**

This paper proposes an interesting problem of online estimation in which the mean of stochastic samples abruptly changes at multiple time-steps. The authors claim that this method achieves nearly minimax-optimal performance without requiring detectability assumptions that are present in prior works. However, some issues arise when connecting the ATC algorithm with the prior works with detectability assumptions, which are not addressed in the paper and will be explained later in this review.

**Strength**
1. The ATC algorithm shows good performance both in synthetic dataset and in real-world dataset. The missed change points do not lead to exogenous confounding.
2. An information-theoretic lower bound is established, which is not present in prior works.
3. The ATC algorithm does not require any prior knowledge on the number of change points nor the horizon to operate, indicating its practicality when applied in real-word scenarios.

**Weakness**
1. Discrepancy between prior works with detectability assumption: A major issue in this work is that the "minimax regret" is different from the that in the prior works with detectability assumptions. In (Padakandla et al., 2020; Besson et al., 2022), the dynamic cumulative regret is the **summation of the difference between the mean of the reward associated with the optimal policy and that associated with the proposed policy**. However, in this work, the dynamic cumulative regret is the **summation of the squared error between the estimate and the true mean of the current sample**. This discrepancy in the regret metrics makes the third strength of this work questionable, as the exemption from detectability assumptions could potentially be a byproduct of using a different regret notion. There might exist an explanation to bridge this discrepancy in the regret metric, but the author did not provide any discussion on this issue.
2. Soundness of theoretical results: In Lemma 3.1, the authors **do not specify whether the upper bound on the difference between SNRs holds deterministically or with high probability**. In the proof of Lemma 3.1 in Appendix A.6, it appears that the decomposition in App. A.4.3 that the author applied on line 1901 is stochastic: The event $E_{t}(r(j))$ is included in $\mathcal{E}(r(j))\cup H_{t}(r(j))$, indicating that $E_{t}(r(j))$ implies $H_{t}(r(j))$ when $\mathcal{E}(r(j))$ fails to hold. Therefore, in order to use the implication on line 1901, you need to conditioned on the event $\bar{\mathcal{E}}(r(j))$, which happens with high probability due to some concentration inequality for sub-Gaussian samples. The authors do not demonstrate what the probability of Lemma 3.1 holding is, and thereby posing some issues to the soundness of the theoretical results.
3. Inaccuracy in references: On line 103, there are no detectability assumptions in (Huang & Veeravalli, 2025). In fact, this is a QCD work with only one change point in the environment. The work with the detectability assumption is probably [1]. In addition, there are no detectability assumptions in (Auer et al., 2019), although the authors of (Besson et al., 2022) show that the proposed method in (Auer et al., 2019) performs poorly on synthetic data.
4. Minor issues with experiments: In Figure 3(c), it would be helpful if the authors plot the lower bound of the cumulative dynamic regret and compare the gap between the lower bound and the regret of the ATC algorithm.
5. Inaccuracy in the discussion: In Section 4.3, the reason for the $\sqrt{ST}$ regret in non-stationary online learning is also caused by the fact that the learner can only observe the reward associated with one action at each time step, while the observed action might not undergoes a change at each change point. The difficulty in these non-stationary online learning problems comes from the fact that the learner do not know which suboptimal action to sample to observe the change, and the suboptimal action cannot be chosen too frequently in case of incurring huge regret.
6. Efficient variant: It would be helpful to mention that Lemma 3.1 and Theorem 4.1 are for the original ATC, not for the computationally efficient variant. The authors can mention that this variant is proposed for experiments in Section 5.
7. Some issues with writing: When introducing acronyms (e.g., CUSUM, NAB), please explain what they stand for.

[1] Huang, Yu-Han, et al. "Detection Augmented Bandit Procedures for Piecewise Stationary MABs: A Modular Approach." arXiv preprint arXiv:2501.01291 (2025).

---

> ### Author Rebuttal · Authors · 2026-03-30
>
> We thank the reviewer for the detailed and informed comments. We believe that several of the concerns reflect a difference in perspective between our learning-theoretic formulation and the bandit settings. We clarify this distinction and respond to each point.
>
> **1. On the discrepancy and relation to bandit/RL**:
> Our paper does not study bandit action selection. It considers a sequential mean-tracking problem with noisy observations under abrupt changes. Since the goal is to accurately estimate the current mean, we evaluate performance using the most natural metric: the expected cumulative squared error between the estimate and the underlying mean. This metric appears in several related areas, e.g., adaptive design in multiperiod control [Lai and Robbins, 1979].
> At the same time, our formulation connects naturally to online learning. The learner’s output $\hat \mu_t$ corresponds to an action $x_t \in \mathbb R$, the loss function is $\ell_t(x_t) = (x_t -\mu_t)^2$, and the dynamic regret $E[\sum_{t=1}^T \ell_t(x_t)] - \sum_{t=1}^T \inf_{x} \ell_t(x) $ reduces to $E[\sum_t (x_t - \mu_t)^2]$ as in [Baby and Wang (2019)]. The piecewise-stationary environment is common in the bandit literature, motivating our references to that line of work.
> To avoid confusion, we will refer to the objective as expected cumulative squared error.
>
> Although our objective differs from the bandit setting, it is not weaker. It isolates the complexity due only to multiple change points by removing additional considerations. We view this work as stand-alone, providing a characterization of the intrinsic complexity of the multiple change-point problem. The absence of detectability assumptions is central: missed detections are no longer "rare events", requiring new analysis and leading to confounding.
> We therefore believe that the third strength of the work (upper bound) is well-founded and firmly established under an appropriate regret criterion.
>
> **2. Designing sampling algorithms**: Our analysis can inform decision-making settings as follows: it may be suboptimal to devote excessive effort to detecting every change point. Instead, our analysis suggests that the key quantity to control is the level of confounding and its impact on regret. As long as this remains sufficiently small, the algorithm may still be able to identify near-optimal actions, even without detectability assumptions. Making this connection fully rigorous requires further work, as the need for exploration introduces additional challenges. We will clarify the challenges of extending our analysis to the bandit setting.
>
> **3. Lemma 3.1 and soundness**: The reviewer is correct that Lemma 3.1 should be stated as a high-probability result. For a fixed restart time $r$, the bound holds on the concentration event from Lemma A.2 with probability at least $1-\alpha_r$, where $\alpha_r$ is such that $\sum_r \alpha_r \leq \alpha$ to bound the total failure probability across all restart times. We will revise the statement of Lemma 3.1.
> Note that this is only an error in the statement of the Lemma, and not a flaw in the correctness of the proofs. Moreover, it does not affect the validity of Theorem 4.1, since the theorem establishes an expected regret bound, and its proof separates the deterministic drift from the low-probability deviation events which are accounted through their total probability yielding the $M^2 \alpha$ term in Eq.(15).
> We thank the reviewer for pointing out this important issue.
>
> **Inaccuracy in references**: We cite [Auer et al. (2019)] for its active-adaptive strategy (with detection and restarts). We agree it should not be grouped with detectability-based works and will revise accordingly.
> In our view, [Huang and Veeravalli (2025)] does impose detectability assumptions (in the single change setting). In the unknown distributions setting, it is assumed that the change point $\nu > m \geq 8 \sigma^2 \Delta^{-2} \beta(T, \delta_F)$, i.e. their high-confidence guarantee holds when sufficiently many pre-change observations are available. This is tied to the structure of the GLR, where the information depends on both pre/post-change sample sizes.
> We thank the reviewer for pointing out [1]; our work could help addressing the detectability challenge identified therein, as discussed.
>
> 5. The claim in Section 4.3 is specific to our formulation: under $\ell_1$ loss, multiple change points alone suffice to induce $\sqrt{ST}$-type behavior, *even without the need for exploration*. This yields an interesting contrast with piecewise-stationary bandits, where exploration is a primary driver to the $\sqrt{ST}$ lower bound, as noted by the reviewer.
>
> 4+6+7. We will add the lower bound to Fig.3(c), state that Lemma 3.1 and Theorem 4.1 analyze the original ATC, and expand acronyms at first appearance.
>
> We thank the reviewer and will revise the paper to address these concerns. We hope that our clarifications enable clearer assessment of the paper’s contributions.

---

> > ### Author Rebuttal · Reviewer_5UDN · 2026-04-03
> >
> > After reading the review, the following concerns has remained unaddressed:
> >
> > 1. On the discrepancy and relation to bandit/RL: If you view this as a stand-alone work different from piecewise stationary bandits, then the detectability assumption is not comparable. The authors should cite works with detectability assumption in sequential mean-tracking problem, or they have to explicitly emphasize that they want to see whether the detectability assumption is required under different piecewise stationary problem.
> >
> > 2. Designing sampling algorithms: The possibility of identifying near-optimal actions is unlikely, as the suboptimal actions are not observed every round. In your problem, the stochastic samples are fully observed at each round. However, this is not the case for piecewise stationary bandits, as only the reward samples for one arm is observed every round. Therefore, until the authors proposed a heuristic sampling algorithm that could possibly work under piecewise stationary bandits with some intuition, I am not convinced that this can possibly be extended to bandit problems.
> >
> > All the other concerns are addressed. I appreciate the authors' effort put into the responses, but the motivation in the introduction is not written in the best way, and the authors did not address the problem raised in the introduction. I will keep my score for now and reevaluate my score based on the authors' response.

---

> > > ### Author Response · Authors · 2026-04-03
> > >
> > > We sincerely thank the reviewer for the thoughtful follow-up and for engaging deeply with our rebuttal. The feedback has already been very helpful in improving the paper.
> > >
> > > **On the central objective of the paper**
> > >
> > > We restate the core objective of the paper as outlined in the introduction: to extend quickest change detection (QCD) from the classical single-change setting to the multiple change-point regime. While the single-change setting is well understood, principled extensions to multiple changes remain limited and typically rely on detectability assumptions (e.g., [Maillard, 2019], [Yu et al., 2023]).
> > > Our work asks whether multi-change QCD can be addressed without such assumptions. We formalize this question through a minimax regret criterion, leading to a mean-tracking formulation under abrupt changes. In this regime, information-theoretic limits preclude high-probability detection guarantees. As a result, failure events such as missed detections are no longer necessarily rare and must be treated explicitly in the analysis. This is precisely what gives rise to the confounding phenomenon and drives both the algorithm design and the novel analysis. Therefore, removing detectability assumptions is central to our formulation. In this sense, our upper and lower bounds directly address the question raised in the introduction by characterizing the regret of multiple-change QCD without detectability assumptions.
> > >
> > > **On the relation to piecewise-stationary (PS) bandits**
> > >
> > > We agree that the comparison to PS bandits must be presented more carefully. Because bandits have partial feedback, detectability there is intertwined with exploration and sampling allocation, whereas in our setting the noisy sequence is fully observed. So we should not present the assumptions as directly comparable. Our intent was only to note that detectability assumptions in PS bandits also play a similar technical role to those in QCD, as high-probability detection guarantees are used as a key component in regret analyses.  For instance, in [Besson et al., 2022], high-probability detection guarantees (Lemma 8, under detectability assumptions from Asssumption 4) are a key ingredient in the main upper bound analysis (Theorem 5). **We will revise this part in the introduction and related-work discussion** so that bandits are presented only as context, and better distinguish: (i) our stand-alone statistical objective from (ii) the role detectability plays across different formulations. We again thank the reviewer for pointing out this important issue.
> > >
> > > We fully agree with the reviewer on the difficulty of extending our framework to bandit settings, and we will emphasize this in the revision. Our earlier discussion was meant as heuristic intuition only. We do not claim to solve this problem; rather, we explicitly highlight it as a direction for future work.
> > >
> > > In summary, while we acknowledge the reviewer’s concerns regarding the presentation of detectability assumptions, we hope the clarification above helps convey the main contributions and conceptual insights of our work.

---

### Official Review · Reviewer_PvKG · 2026-03-08

**Soundness:** 2
**Presentation:** 1
**Significance:** 1
**Originality:** 1
**Overall Recommendation:** 3
**Confidence:** 4

**Summary:**

The paper proposes improvement to online change detection targeted to the circumstances where multiple change points occur relatively close to each other in time. The study promises to balance out sensitivity and stability of change detection. The proposed solution extends a classical sequentil change detection algorithm. Balancing sensitivity and stablity comes down to trading off detection accuracy vs. false alarms, which is the classical conandrum in change detection. While the paper is dense in formalisms it is not entirely clear where the expected gain in performance comes from theoretically.

**Compliance With Llm Reviewing Policy:**

Affirmed.

**Key Questions For Authors:**

Why would one whant to reconsile multiple change points happening close to each other? Would it be sufficient to flag the whole region of multiple change points as transitional?
What are the main implications and conclusions resulting from the numerical experiments?

**Limitations:**

Lack of balance: the study is heavy in formalisms and light in experimental analysis. The two parts could be better integrated.

**Strengths And Weaknesses:**

Strengths
-	Historical account
-	Visual explanations
Weaknesses
-	Theoretical expectations regarding the source of performance gain are not entirely clear
-	Motivation for accurately detecting multiple changes happening close to each other is not clear
-	Experimental section feels rushed through, lacks compehensive analysis and interpretations

---

> ### Author Rebuttal · Authors · 2026-03-30
>
> We thank the reviewer for the comments and questions. We believe some of the concerns arise from a difference in perspective regarding the primary goal of the paper. Our contribution is mainly theoretical: we study the multiple change-point problem through a learning-theoretic lens, focusing on the regret induced by changes and the resulting statistical tradeoffs. This perspective leads to sharp and novel upper and lower bound guarantees, formalized in Lemma 3.1 and Theorems 4.1 and 4.2. The simulations in the main paper are therefore designed to validate these theoretical results, while additional empirical analysis is deferred to the appendix due to space constraints. We address the reviewer’s comments and questions below.
>
> **1. Why reconcile multiple nearby change points? Why not simply flag the whole region as transitional?**
>
> This question is central to our contribution. The key point is that, in the multiple change-point setting, not all changes should be treated equally. Our analysis precisely quantifies when a region can be treated as effectively "transient" under the regret objective. Specifically, under our minimax formulation, the effect of a change point is determined not only by how close it is to the previous one, but also by the magnitude of the jump (and the noise level that can mask it). In particular, even a short segment may still have a substantial impact on learning if the mean shift is sufficiently large, whereas a longer segment with a small shift may be relatively unimportant. Thus, the relevant notion is not merely whether changes are "close", but rather the tradeoff between segment length and jump size.
> One of the main contributions of the paper is to make this tradeoff precise. In particular, our lower bound identifies the intrinsic cost of such changes using information-theoretic arguments based on the data processing inequality, while the upper bound shows how the ATC threshold should be chosen to distinguish between transient and significant segments. In that sense, the paper develops a principled way to distinguish between significant and "less important" changes under the regret criterion.
>
> **2. Where does the performance gain come from?**
>
> Unlike classical high-confidence frameworks that treat changes as "detected" or "missed", our analysis quantifies the cost of a change based on its statistical impact. This leads to a selective detection principle: changes with small regret contribution can be safely ignored, while significant ones must be detected quickly. Endogenous confounding plays a key role, missed changes mix regimes and degrade the effective signal-to-noise ratio of future detection. By quantifying this effect, we derive thresholds that balance detection and estimation, yielding near-minimax guarantees.
>
> **3. What are the main conclusions from the numerical experiments?**
>
> Since the paper's contribution is mainly theoretical, the role of the experiments in the main text is therefore focused: to validate the
> theoretical results and to demonstrate that the algorithm is applicable to real-world data.  That said, the appendix includes a substantially more comprehensive experimental study with several important insights, which we will emphasize more clearly in the revised main text. In particular:
> (i) Appendix A.1.2 studies dense-change regimes and empirically characterizes the range of change-point densities (relative to the
>   horizon) under which the algorithm no longer achieves sublinear regret, in line with the theory.
> (ii) Appendix A.1.4 examines long-horizon behavior and highlights the anytime nature of the method. In particular, it shows that while constant thresholds may perform better in some finite known-horizon regimes, the ATC threshold becomes preferable when the horizon is long  or unknown.
> (iii) Appendix A.1.5 studies sensitivity to variance misspecification.
> (iv) Appendix A.1.6 constructs adversarial environments that illustrate more concretely which combinations of segment length and jump size maximize the regret.
> We agree that these insights should be better surfaced in the main paper. In the revision, we will strengthen the discussion of the
> experimental conclusions in the main text and explicitly point the reader to the broader experimental analysis in the appendix.
>
> **4. On the motivation and significance**
>
> We believe the significance of the paper may have been underappreciated in the review. The paper introduces a new viewpoint on the multiple change-point problem: rather than imposing detectability assumptions and focusing only on high-confidence detection, it studies the minimax cost of learning under multiple changes, including the unavoidable effect of missed detections. This leads to a new characterization of the problem, a new algorithmic principle based on selective detection, and matching upper/lower bounds up to logarithmic factors.
>
> We thank the reviewer again for the feedback.

---

> > ### Author Rebuttal · Reviewer_PvKG · 2026-04-03
> >
> > Thanks for your comments. Indeed, the main differences are in the perspectives which are not easily resolvable by a revision. I'm still not convinced about the significance of the problem setting and the contributions of the paper.

---

### Official Review · Reviewer_9ax2 · 2026-03-12

**Soundness:** 3
**Presentation:** 2
**Significance:** 4
**Originality:** 3
**Overall Recommendation:** 5
**Confidence:** 3

**Summary:**

This paper studies sequential mean tracking in piecewise-stationary environments where the data distribution can change an unknown number of times abruptly. The paper identifies a key weakness of traditional high-confidence change-point detection methods in multi-shift settings, termed endogenous confounding: when a change point is missed, the algorithm mixes pre- and post-change samples, weakening the signal for future statistical tests and leading to persistent tracking errors and large regret.

To address this issue, the paper proposes the Anytime Tracking CUSUM (ATC) algorithm, an online method that does not require a predefined time horizon. ATC uses a gradually increasing detection threshold, creating a selective detection mechanism that ignores small shifts that are hard to detect while still responding quickly to larger changes.

This work provides theoretical guarantees regarding ATC's dynamic regret and also derives an information-theoretic minimax lower bound using Kullback–Leibler data-processing arguments, demonstrating that ATC is nearly minimax optimal. Experiments on both synthetic datasets and the NAB CPU utilization benchmark confirm that ATC outperforms sliding-window and exponentially discounted baseline methods.

**Compliance With Llm Reviewing Policy:**

Affirmed.

**Final Justification:**

My concerns have been resolved. I will keep the score as it is.

**Key Questions For Authors:**

1. Could the authors include a brief comparison between ATC’s data-driven adaptive reset mechanism and periodic reset strategies ([1], [2], [3]), which deterministically clear memory to avoid endogenous confounding? Such a discussion would better position ATC within the broader literature.

- For example, periodic-reset approaches like PROPO [1] and R-MOSS [2] achieve near-optimal and order-optimal bounds in MDPs and MABs, respectively, while similar ideas are used in chaotic applied settings such as resilient swarm robotics (e.g., the AOL framework [3]) to maintain plasticity after undetected sensor shifts.
- How do the theoretical and practical trade-offs between these approaches compare?

2. Since performance depends on the variance parameter $ \sigma $, how easily could ATC incorporate an online variance estimator in settings where the variance is unknown or time-varying? Would this affect the theoretical guarantees?

**References**:
- [1] Zhong, Han, Zhuoran Yang, Zhaoran Wang, and Csaba Szepesvári. "Optimistic policy optimization is provably efficient in non-stationary MDPs." arXiv preprint arXiv:2110.08984 (2021).
- [2] Wei, Lai, and Vaibhav Srivastava. "Nonstationary stochastic bandits: UCB policies and minimax regret." IEEE Open Journal of Control Systems 3 (2024): 128-142.
- [3] Gupta, Shubhankar, Saksham Sharma, and Suresh Sundaram. "Reward-based Autonomous Online Learning Framework for Resilient Cooperative Target Monitoring using a Swarm of Robots." Transactions on Machine Learning Research (2025).

**Limitations:**

yes

**Strengths And Weaknesses:**

1. Soundness:

- Strong technical rigor with a clear formulation of endogenous confounding.
- Careful regret decomposition, separating variance and bias contributions.
- Theoretical results avoid restrictive assumptions such as minimum shift size or spacing between change points, improving over prior change detection literature.
- Well-designed experiments that validate the predicted logarithmic regret behavior.
- Performance is sensitive to the misspecification of the variance parameter $ \sigma $ ; underestimating it can increase the regret in practice.

2. Presentation:

- The paper is very well written and logically organized, with a clear progression from intuition to formal analysis.
- Algorithm description is clear, and detailed proofs are moved to the appendix, keeping the main text readable.
- The literature review focuses mainly on CUSUM and change-point detection; a broader discussion of other non-stationary adaptation approaches (e.g., periodic reset methods) would improve context.

3. Significance:

- Addresses an important and widely relevant problem in machine learning and signal processing.
- The framework could influence research in online tracking, time-series forecasting, and non-stationary bandits.

4. Originality:

- Introduces the novel concept of endogenous confounding in the context of dynamic regret minimization.
- Provides a rigorous theoretical framework to analyze the impact of missed change detections.
- Proposes a horizon-free, time-varying threshold that explicitly controls the endogenous confounding effect.

---

> ### Author Rebuttal · Authors · 2026-03-30
>
> We thank the reviewer for the thoughtful and supportive review, and for the positive evaluation of the paper. We also appreciate these two questions, as they help us better position the contribution relative to the broader non-stationary learning literature and highlight a natural direction for extending the framework. We will add a brief discussion of both points in the revision.
>
> 1. **Comparison with periodic reset and related passive adaptation approaches**
>
> We agree that periodic-reset, sliding-window, and discounting methods are important and powerful approaches for non-stationary online learning, and we appreciate the reviewer for pointing out these connections. More
> broadly, these methods belong to the class of passively adaptive strategies, where forgetting is enforced according to a predetermined schedule or decay rule, rather than being triggered by the data.
>
> The key distinction between ATC and the periodic resets, sliding windows, and discounting methods is that they require specifying a reset frequency, window length, or discount factor in advance. In the existing literature, including [1] and [2], these tuning parameters are typically chosen as functions of problem-dependent quantities such as the horizon $T$ or the variation budget (which plays a role analogous to the number of changes $S$ in our formulation). In [3], the reset time is treated as a hyperparameter. When such quantities are known or can be tuned properly, these methods can indeed achieve strong and even nearly-optimal guarantees.
>
> However, when they are unknown or misspecified, it is less clear how to tune them optimally.  This is precisely the regime in which we expect ATC to be superior: it is horizon-free, does not require prior knowledge of $T$ or $S$, and adapts directly to the observed data. In our experiments, we already compare ATC against sliding-window and discounted-mean baselines (Figures 4 and 5b), with their parameters tuned using an offline dataset for fairness rather than chosen using oracle knowledge of $T$ or $S$. In these comparisons, ATC performs favorably.
> At the same time, we do not claim that ATC necessarily outperforms such passive methods when the latter are optimally tuned with access to $T$ and  $S$. In fact, in the first discussion point of Section 4.3, we explicitly note that if such prior knowledge is available, then one may be able to design restart schemes with even sharper regret guarantees. So we view the comparison as a tradeoff:  oracle-tuned passive methods can be very strong when problem parameters
> are known, whereas ATC is more robust when they are not.
> We agree that this is an important positioning point, and we will expand this discussion in the revised paper.
>
> 2. **Unknown or time-varying variance**
>
> We thank the reviewer for raising this point. Indeed, the current ATC analysis assumes knowledge of the variance proxy $\sigma$, and understanding how to relax this assumption is an important direction. On the empirical side, we already include a sensitivity study under variance misspecification (Figure 7b), which shows that ATC is reasonably robust to moderate overestimation but can suffer when $\sigma$ is substantially underestimated. This suggests that incorporating variance estimation would be practically valuable.
>
> A natural extension would be to combine ATC with an online estimator of the variance and plug the resulting estimate into the threshold. At a high level, the key challenge in the analysis would be to control the additional error introduced by estimating $\sigma$, especially in the early stages where the estimator may be inaccurate. Since the variance is constant in our current formulation, one can expect the estimate to improve over time, so the main issue is to quantify how this transient estimation error propagates through the detection threshold and regret decomposition.
>
> We have not developed this extension fully in the current paper, so we prefer not to make a formal claim here. Our expectation is that finite-time concentration bounds for the variance estimator could be incorporated into the analysis, leading to additional lower-order terms in the regret. Whether the same logarithmic regret order can still be maintained is an interesting open question, and we will add this explicitly as a future-work discussion point in the revision.
>
> Thank you again for these helpful suggestions. We believe both discussions will strengthen the paper and improve its positioning.

---

> > ### Author Rebuttal · Reviewer_9ax2 · 2026-04-02
> >
> > My concerns have been resolved. I will keep the score as it is.

---

### Official Review · Reviewer_p5ko · 2026-03-13

**Soundness:** 3
**Presentation:** 3
**Significance:** 3
**Originality:** 3
**Overall Recommendation:** 5
**Confidence:** 3

**Summary:**

This paper studies an online learning problem in piecewise stationary environments with multiple change points, particularly under the online tracking setting with a dynamic regret formulation. This paper proposes a new class of learning algorithms called Anytime Tracking CUSUM (ATC), and derives the upper and lower bounds to the regret.

**Compliance With Llm Reviewing Policy:**

Affirmed.

**Final Justification:**

I kept my original evaluation.

**Key Questions For Authors:**

The current setup and theoretical analysis focus on mean value tracking in piecewise-stationary environments. How does the framework generalize to other types of distributional change, such as covariate shift or variance change in vector-valued observations?

How does the regret formulation extend beyond piecewise-stationary environments, e.g., to gradually changing means? Since the regret formulation itself does not explicitly assume piecewise stationarity, it may be applicable in more general non-stationary settings. A brief discussion of such possible extensions would broaden the scope of the paper.

**Limitations:**

yes

**Strengths And Weaknesses:**

Strengths
- The formulation of multiple change-point detection as regret minimization is a novel and practical framework, with great applicability in many real-world scenarios. Instead of focusing solely on minimizing the detection delay when there is a single change-point, the authors tackle a more challenging problem by analyzing the overall regret when there might be multiple change-points and missed detections, without assuming prior knowledge of the time horizon and number of change-points.
- The authors made solid theoretical contributions to the regret analysis, including both the upper and lower bound to the regret.
- The algorithm is simple and efficient (with geometric offsets) to implement in practice.

Weaknesses
- The current setup and theoretical analysis mainly focus on the mean tracking problem and rely on the independence assumptions. Though this is acceptable for theoretical considerations, it may limit its practical usage in real-world scenarios.
- The numerical experiments are a bit limited, with only an univariate synthetic example and one real dataset, and there is no comparison with baseline methods for multiple change-point detection.

---

> ### Author Rebuttal · Authors · 2026-03-30
>
> Thank you for the thoughtful review and for the positive evaluation of our paper. We also appreciate these two questions, since they point to natural extensions for the  present framework.  We will add a brief discussion of these directions in the revision. Below, we provide a clear and detailed response to the question raised by the reviewer.
>
> 1. **Extension to broader statistical settings**
>
> We chose the mean-tracking  since it provides a clean and canonical setting in which one can isolate the intrinsic statistical difficulty imposed by multiple change points themselves, without additional modeling layers. Our lower bound, together with the endogenous confounding phenomenon that emerges in this setting, highlights that even this canonical formulation is nontrivial.
>
> We agree that the present formulation is simple, by design. A natural extension is to  to model the  observations as IID according to the exponential family distribution $f_\theta(x)=h(x)\exp\left(\langle \theta, T(x) \rangle - \Psi(\theta)\right)$, where the parameter $\theta \in \Theta \subseteq \mathbb{R}^d$, $T(x) \in \mathbb{R}^d$ is the sufficient statistic, and $\Psi(\theta)$ is the log-partition function.  In such a setting, the analogue of “tracking the mean” is to track the expectation  parameter $\eta(\theta) \triangleq \mathbb{E}_\theta[T(X)] = \nabla \Psi(\theta)$. A natural ATC-type extension would therefore replace sample averages of $X$ by sample averages of the sufficient statistic $T(X)$, and compare pre- and post-change empirical sufficient statistics. The main technical ingredients that make such an extension plausible are the following. First, if $(T(X) - \eta(\theta))$ satisfies a suitable uniform concentration bound over the parameter class of interest (for example, sub-Gaussian concentration), then one can expect an analogue of our
> confidence bound for the detection statistic. Second, if $\Psi$ has suitable local curvature properties (e.g., strong convexity / smoothness on the relevant parameter region), then the KL divergence is locally comparable to a quadratic form, which suggests that a similar bias-variance-detection tradeoff underlying our analysis should remain valid, up to geometry-dependent constants.
>
> This would cover a broader class of changes than scalar mean shifts, including, for example, variance changes or other parametric changes in vector-valued observations. As one concrete example, a Gaussian model with unknown mean and variance can be written as an exponential family with sufficient statistic $T(x) = (x,x^2)$. We have not developed this extension fully in the current paper, so we prefer not to claim a complete theorem here, but we believe it is a promising and technically accessible direction.
>
> 2. **Extension beyond piecewise-stationary environments**
>
> We also thank the reviewer for this question. While the current paper focuses on abrupt changes as our goal is to understand the complexity of the multiple change point problem, we agree that the regret formulation itself is more general and may be useful beyond piecewise-stationary models.
>
> In particular, some of the main insights of the paper appear relevant to gradually drifting environments as well, as suggested by the reviewer. At a high level, our results suggest that the  key tradeoff is between retaining past data to reduce variance, and discarding outdated data to control bias. Under gradual drift, as long as the cumulative drift over a time window remains below the statistical error level of estimation, reusing past samples is beneficial. Once the drift becomes large enough relative to this estimation scale, the older samples contribute more bias than variance reduction, and a restart becomes preferable. From this viewpoint, an ATC-type procedure may be interpreted not only as a change-point detector, but more broadly as an adaptive segmentation rule driven by statistical significance.
>
> We expect that such settings would lead to a different regret characterization than the logarithmic-in-$T$ dependence established here for piecewise-stationary means, since now the environment itself may contribute approximation error even in the absence of abrupt jumps. We therefore view this as an interesting extension rather than a direct corollary of our current results.  We will add a short discussion of this point in the revised paper.
>
> Thank you again for highlighting these directions. We agree that including such a discussion would broaden the scope of the paper and better clarify the potential reach of the framework.

---

> > ### Author Rebuttal · Reviewer_p5ko · 2026-04-04
> >
> > I will keep my score.

---

### Decision · Program_Chairs · 2026-04-30

**Decision:**

Accept (regular)

**Comment:**

This paper studies a stand-alone learning-theoretic formulation of mean tracking under multiple change points and proposes the Anytime Tracking CUSUM (ATC) algorithm. Its main strengths are the formulation of endogenous confounding, the horizon-free ATC procedure, and a fairly complete theoretical treatment through upper and lower bounds that are close up to logarithmic factors. The discussion also made it clearer that the paper is best evaluated as a stand-alone multiple-change-point mean-tracking work, rather than through direct comparison with piecewise-stationary bandit formulations.

The internal discussion also helped clarify one substantive concern about whether the main results might already follow from prior work, and that concern was softened in the later discussion, which increased my confidence in the paper’s core contribution. While one reviewer remained unconvinced, I read that negative assessment as reflecting a more skeptical view of the broader significance of the formulation and the limited empirical scope, rather than a concrete issue with the technical soundness of the work. Taking the reviews and follow-up discussion together, I am supportive of the paper overall. For the final version, however, the introduction and overall positioning should be revised so that the paper is clearly presented as a stand-alone multiple-change-point mean-tracking problem, with comparisons to piecewise-stationary bandits used only for context, rather than suggesting that the two settings are directly comparable.